# TOOLDIAL: MULTI-TURN DIALOGUE GENERATION METHOD FOR TOOL-AUGMENTED LANGUAGE MODELS

**Jeonghoon Shim**[1], **Gyuhyeon Seo**[1], **Cheongsu Lim**[2], **Yohan Jo**[1*]
[1]Graduate School of Data Science, Seoul National University
[2]Department of Industrial and Management Engineering, Korea University
jhshim98@snu.ac.kr

## ABSTRACT

Tool-Augmented Language Models (TALMs) leverage external APIs to answer user queries across various domains. However, existing benchmark datasets for TALM research often feature simplistic dialogues that do not reflect real-world scenarios, such as the need for models to ask clarifying questions or proactively call additional APIs when essential information is missing. To address these limitations, we construct and release ToolDial, a dataset comprising 11,111 multi-turn dialogues, with an average of 8.95 turns per dialogue, based on APIs from RapidAPI. ToolDial has two key characteristics. First, the dialogues incorporate 16 user and system actions (e.g., "Request", "Clarify", "Fail inform") to capture the rich dynamics of real-world interactions. Second, we simulate dialogues where the system requests necessary information from the user based on API documentation and seeks additional APIs if the user fails to provide the required information. To facilitate this process, we introduce a method for generating an API graph that represents input and output compatibility between APIs. Using ToolDial, we evaluate a suite of language models on their ability to predict correct actions and extract input parameter values for API calls from the dialogue history. Modern language models achieve accuracy scores below 70%, indicating substantial room for improvement. We release our dataset and code at https://github.com/holi-lab/ToolDial.

## 1 INTRODUCTION

A Tool-Augmented Language Model (TALM) is a language model designed to select and call appropriate tools (usually APIs) while interacting with the user to answer the user's query. By leveraging external tools, the TALM can conduct complex tasks beyond its parametric knowledge and adapt its actions based on API results. Recent TALM benchmarks mostly feature single-turn interactions (Qin et al., 2023; Tang et al., 2023) with a primary focus on improving tool selection and reasoning capabilities to address complex user queries within a single turn. However, such interactions do not reflect real-world scenarios where the TALM should request additional information from the user or the user clarifies their intent. Even in studies that involve multi-turn interactions (Li et al., 2023), dialogues tend to be short and limited to scenarios where the TALM asks the user for more details. The lack of richer datasets that reflect complex user-system interactions makes it difficult to accurately assess the ability of modern language models to handle challenging tool use scenarios in the wild, such as when the system identifies and requests information from the user based on available APIs, or when the user cannot provide requested information, requiring the model to call additional APIs to obtain the information.

To address this issue, we present a new dataset named ToolDial, which consists of multi-turn dialogues between the user and TALM based on APIs from RapidAPI[1]. The main focus of our dataset is to simulate dialogues where multiple APIs should be called in sequence (e.g., due to the user failing to provide information that is needed to call the main API) and where the user and the TALM can take diverse actions (16 total), such as clarifying the user's intent or handling the user's failure

---

*  Corresponding author.
[1]https://rapidapi.com/hub

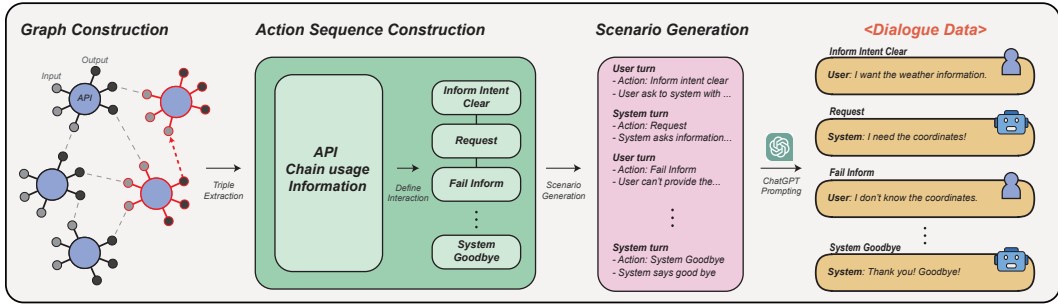

Figure 1: Overall structure of ToolDial. This represents the whole pipeline of our method.

to provide requested information. To that end, our data generation pipeline consists of four steps, as shown in Figure 1. First, to facilitate selecting two APIs that should be called in sequence, we construct an API graph where nodes are APIs and edges between two APIs indicate that one API's output can be used as input for the other API (§3.1). Second, to simulate rich dynamics between the user and TALM, we define 16 types of user and system actions informed by the literature of task-oriented dialogue systems and compile 23 plausible sequences of actions that are likely to occur in dialogues (e.g., Inform intent clear → Retriever call → Request → Fail inform) (§3.2). Third, to generate each dialogue, we select a pair of APIs from the API graph and choose a sequence of actions that serves as a skeleton. Based on this, we enrich the skeleton by incorporating additional dialogue state information for each turn, such as the input parameters of the APIs informed by the user (§3.3). Fourth, we convert the augmented action sequence into natural utterances to complete a dialogue (§3.4). As a result, ToolDial contains 11,111 dialogues with an average of 8.95 turns per dialogue.

Based on ToolDial, we designed three evaluation tasks to assess a suite of language models in their ability to use tool. Specifically, we evaluated their ability (1) to predict appropriate actions to progress toward answering the user query, (2) to choose the correct API and predict dialogue states (i.e., extracting user-informed values for API inputs), and (3) to generate responses faithful to API outputs. We found that GPT-based models struggle with dialogue state prediction, and their performance declines as the dialogue length increases. Additionally, these models perform poorly at predicting next actions, particularly struggling with requesting input parameters and asking clarifying questions. For smaller Llama models, they generally underperform compared to GPT-based models, but fine-tuning on our dataset significantly improved the performance of each task. Notably, it led to substantial improvements in many actions that GPT models struggled with. Our experiments suggest that ToolDial can be a valuable resource for both assessing and improving TALMs in complex multi-turn interactions with users.

The main contributions of our work are summarized as follows:

- We generate and release ToolDial, a dataset consisting of dialogues that reflect real-world interactions between the user and a TALM, encompassing 16 user and system actions.
- We present a framework for creating a large-scale and multi-turn dialogue benchmark using an API graph and GPT-4o with minimal human effort.
- We provide insights into the abilities of various language models to answer user queries while interacting with the user across multiple turns and using external APIs.

## 2 RELATED WORKS

**Tool Augmented Language Models**    Table 1 compares our dataset with existing benchmarks. Recent research on TALM has evolved toward investigating how to effectively select tools and determine which reasoning steps are beneficial for solving complex problems (Yao et al., 2023; Schick et al., 2023; Shen et al., 2023; Qin et al., 2023; Patil et al., 2023; Tang et al., 2023). Similar to our work, ToolNet (Liu et al., 2024) leverages an API graph, but this graph connects APIs that are called back-to-back in dialogues without considering the compatibility of the input and output of APIs. Most existing datasets contain single-turn dialogues between the user and a TALM. For instance,

Table 1: Comparison between ToolDial and other TALM datasets. We derived the number of actions based on how many action types occur in each dataset with our action taxonomy as a reference.

| Resource | ToolDial | ToolBench | API-Bank | ToolAlpaca |
|---|---|---|---|---|
| Real-world API? | ✓ | ✓ | ✓ | ✗ |
| Multi-turn Scenario? | ✓ | ✗ | ✓ | ✗ |
| Multi-tool Scenario? | ✓ | ✓ | ✓ | ✗ |
| Multi-step Reasoning? | ✓ | ✓ | ✓ | ✗ |
| Situation Complexity? | ✓ | ✗ | ✗ | ✗ |
| Number of Actions | **16** | 3 | 7 | 3 |
| Number of Dialogues | **11,111** | 188,304 | 6,860 | 4,889 |
| Avg. Turn per Dialogue | **8.95** | 2 | 2.84 | 2 |

TaskBench (Shen et al., 2024) attempted to construct graphs by matching API inputs and outputs and generating user queries that can be solved using API chains. However, they did not propose a method for graph construction, and focused solely on inferring the sequence of APIs required to solve a user query in a single turn rather than through a multi-turn dialogue. Although API-Bank (Li et al., 2023) contains multi-turn interactions, the number of turns in each dialogue is limited (2.84 on average), and the interactions are relatively simplistic. ToolTalk (Farn & Shin, 2023) also reflects some degree of multi-turn interactions (6.44 on average), but it relies on dialogue generation using human annotators, resulting in only a small amount of data (a total of 78 dialogues).

**Task-Oriented Dialogue System** A task-oriented dialogue (TOD) system is a goal-oriented dialogue system that processes user queries, understands the intent, and provides answers based on database searches or tool calls. Representative datasets for TOD include MultiWOZ (Budzianowski et al., 2020) and Schema-Guided Dialogue (SGD) (Rastogi et al., 2020). MultiWOZ is a multi-turn dialogue dataset generated by human annotators, which reflects the interactions between users and the system. Additionally, the annotations of dialogue states allow for the evaluation of a system's ability to track dialogue states. Similarly, the SGD dataset features multi-turn interactions. Notably, the way SGD was generated shares similarities with our data generation method, particularly in that an action sequence is chosen first for each dialogue, and then utterances are generated. However, unlike our work, the dialogues in SGD do not reflect difficult situations that a real-world tool agent may face, as SGD utilizes a limited number of APIs and there are no scenarios where the user fails to provide the necessary information for an API call. The literature on TOD offers useful concepts such as dialogue state tracking (Jacqmin et al., 2022) and rich taxonomies of user and system actions that occur in interactions with real-world agents. There have also been attempts to transfer TOD datasets into TALM-style data (Moghe et al., 2024). We designed the ToolDial dataset by referencing representative benchmarks in TOD (e.g., the format of dialogue states in MultiWOZ and action types in SGD).

## 3 ToolDial

The dialogues in ToolDial are generated to reflect complex interactions between the user and system in realistic situations involving chained API usage (i.e., the output of one API is used as the input for another API). To achieve this, we follow four steps, as shown in Figure 1. First, we construct an API graph by connecting the input and output entities of APIs (§3.1). This graph plays a critical role in selecting interdependent APIs to be used for each dialogue. Second, we define 16 types of user and system actions to capture the complex dynamics in interactions with tool agents. Based on these actions, we create 23 plausible action sequences that are likely to occur in dialogues (§3.2). Third, to generate a dialogue, we choose a pair of APIs from the API graph, select an action sequence, and augment it with concrete dialogue states that track the collection of input parameters for the APIs (§3.3). Lastly, we generate utterances that reflect the augmented action sequence using GPT-4o (§3.4). These processes are carried out with minimal human effort.

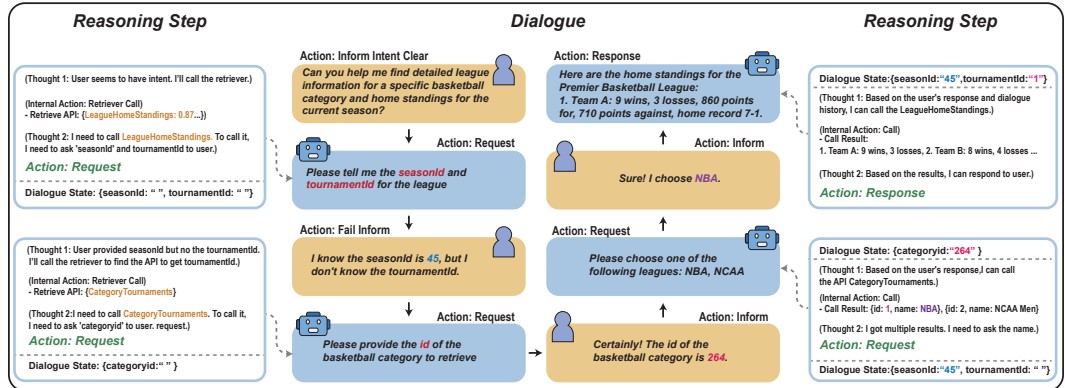

Figure 2: An example dialogue from ToolDial. This illustrates the user and TALM actions for each turn, along with corresponding utterances. It also shows the reasoning steps TALM undergoes, including API calls and retriever calls, before asking or responding to the user.

### 3.1 GRAPH CONSTRUCTION

**Motivation** To simulate dialogues where APIs should be called in sequence to fulfill the user's needs (e.g., the user fails to provide a necessary argument for an API, and thus the system should proactively find and run another API that can provide it), it is necessary to identify which API's output can be used as the input for another API (i.e., API chaining). To facilitate this, we construct an API graph where APIs from RapidAPI are represented as nodes, and two APIs are connected by an edge if one API's output can be used as the input for the other API. Eventually, this API graph will be used in dialogue generation, allowing us to easily select compatible APIs to be called in sequence.

**Settings** To determine whether to build an edge between two APIs, we used the names and descriptions of their input and output entities from the API documentation on RapidAPI. However, these input and output entities often had generic names (e.g., 'id'), and their descriptions did not sufficiently explain their meanings. To address this, we augmented the descriptions using GPT-4o-mini, incorporating the API documentation and instructions (A.1). To replace generic names with more descriptive and informative identifiers, we summarized the augmented description into a 5- to 7-word phrase. Additionally, we extracted up to 4 keywords from each API's description to represent its functionality, ensuring that APIs from vastly different domains were not connected during edge construction (A.2).

**Edge Construction** Using the keywords of APIs, along with the names and descriptions of their input and output entities, we established three criteria for constructing edges $Edge$ based on their similarities. This process is formalized in Equation 1.

$$Edge = \begin{cases} 1, & \text{if } emb(d_o, d_i) > t_d \land emb(d_o + k_o, d_i + k_i) > t_k \land \text{LCS}(n_o, n_i) > t_l \\ 0, & \text{otherwise} \end{cases} \quad (1)$$

where $i$ and $o$ represent the input and output entities, respectively. $d$, $k$, $n$, and $d + k$ denote the description, keywords, name, and the concatenation of keywords and description, respectively. $emb$ is the embedding of a description obtained from the S-BERT model all-mpnet-base-v2 (Reimers & Gurevych, 2019). $LCS$ stands for the longest common subsequence (Hirschberg, 1977). $t$ represents the threshold for each criterion. With the embedding similarity between $d_i$ and $d_o$ and the longest common subsequence similarity between $n_i$ and $n_o$, we aimed to match input and output entities that exactly correspond to each other. Furthermore, by considering the embedding similarity between $d_i + k_i$ and $d_o + k_o$, we ensured that entities from vastly different domains were not incorrectly matched. As a result, we constructed 4,857 edges from 500 million edge candidates (4,474 × 4,474 API pairs, with each pair averaging 25 edge candidates).

**Edge Evaluation**    To verify the edges in the constructed graph, we designed an automated evaluation metric to classify whether each edge was valid (see the examples of mismatched edges in A.11). Directly calling the API would be the most reliable method for validating edges, but it requires a substantial amount of time and cost and suffers from non-executable APIs in RapidAPI. To address this, we utilized StableToolBench (Guo et al., 2024), an API simulator based on large language models. StableToolBench can generate API outputs similar to real API calls, allowing us to validate edges in a similar way to actual API calls. However, StableToolBench also has some issues; for example, the outputs of the same API have different formats upon multiple calls. We fixed such issues by augmenting StableToolBench with additional information from API documentation. We sampled 200 edges from our API graph and measured the Matthews Correlation Coefficient (Matthews, 1975) against human evaluations, which resulted in a score of 0.868. This score indicates a strong correlation between the evaluation metric and human judgment. For the 4,857 constructed edges, the precision (the proportion of valid edges among constructed edges) was 70.9%. Next, to estimate the number of missing edges, we measured Negative Predictive Value (the proportion of invalid edges among non-constructed edges). Since the graph contained too many unconstructed edges (i.e., no connection between APIs), we sampled 5,501 pairs of input and output entities that were not connected. The NPV score was 95.0%, indicating that among the candidates that could form edges, the proportion missing was small. These results indicate that our constructed graph covers most valid edges at the expense of 30% invalid edges. For dialogue generation, we discarded the invalid edges in the subsequent steps.

## 3.2    ACTION SEQUENCES

**Motivation**    In dialogue systems, an action refers to a dialogue act representing a specific behavior taken by the user or system during a conversation (e.g., "request information", "deny suggestion", etc.). A taxonomy of user and system actions allows a dialogue system to manage dialogue flow effectively, by focusing on high-level behaviors before generating utterances and providing interpretability. We compile a taxonomy that covers a wide range of actions occurring in user-system interactions so that the generated dialogues and trained systems reflect the complexity of the real world. To generate a dialogue in the next step, we will first choose a plausible sequence of actions (i.e., dialogue flow) as a skeleton before generating utterances (a similar approach was adopted in SGD (Rastogi et al., 2020)).

**Definition of Actions**    We define a total of 16 actions that the user and system can take. User actions include three types of intent expressions: "Inform intent clear" (an unambiguous query that can specify the correct API), "Inform intent clear add" (an unambiguous query along with one additional input entity of the corresponding API), and "Inform intent vague" (an ambiguous query). Additionally, "Inform" and "Fail inform" refer to the success and failure, respectively, of providing an API's input entities requested by the system. With "Affirm" and "Negate", the user can accept or reject the system's suggestions.

System actions include "Request", which asks the user for information, and "Response", which provides an answer to the user's query. When the user's query is ambiguous, the system may take actions such as "Clarify" or "Suggest" to refine the query. We also define internal system actions such as "Retriever call" and "Call", which occur during the TALM's reasoning steps. The "Retriever call" action retrieves the appropriate API, while "Call" executes the selected API once all input parameters have been obtained from the dialogue history (see the description of actions in A.3).

**Action Sequences**    Based on the predefined actions, we define plausible action sequences (Figure 3). ToolDial is created by combining API pairs from the API graph with action sequences. The types of combinable action sequences depend on whether the APIs in the pair require input parameters and on the form of their outputs (e.g., a single value vs. a list of values).

For example, in Figure 2, the "CategoryTournaments" API outputs "id", which can serve as the input parameter "tournamentId" for the "LeagueHomeStandings" API. Both APIs require input parameters, and "CategoryTournaments" returns a list of "id"s. In this case, the high-level action sequence is as follows:

- Inform intent clear → Retriever call → Request → Fail inform → Retriever call → Request → Inform → Call → Request → Inform → Call → Response.

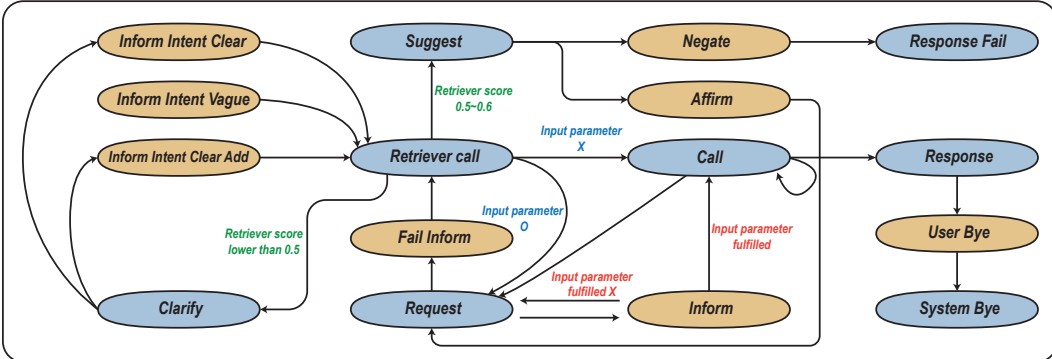

Figure 3: Action graph based on predefined user and system actions. This represents the whole multi turn interaction between user and TALM in our dataset.

There are three "Request" actions in this action sequence. The first one retrieves the input parameters needed to execute "LeagueHomeStandings", the second executes "CategoryTournaments", and the third selects one "id" from the multiple IDs outputted by "CategoryTournaments" (see the 6th turn in Figure 2). If an API required no input parameters or returned a single value instead of a list, there would be at most two "Request" actions, modifying the overall structure of the action sequence.

We also construct different action sequences depending on whether the intent-informing action is "Inform intent clear" or "Inform intent vague". In the latter case, we further distinguish whether it transitions into a "Clarify" or "Suggest" action. Additionally, we design different action sequences based on the user's "Fail inform" action within the same API pair (see details in A.5). The complete set of rules governing action sequences is visualized in Figure 3 (see all types of action sequences in A.6).

## 3.3 SCENARIO INSTRUCTION GENERATION

ToolDial is a collection of task-oriented dialogues where the system utilizes appropriate APIs to achieve the user's goal. When necessary, the system retrieves suitable APIs through an API retriever and collects the required input parameters for API calls through multi-turn interactions. Generating such a dialogue involves simulating a user query, defining dialogue states that specify the required input parameters for APIs provided by the user, and creating utterance instructions that guide utterance generation in the subsequent step.

**User Query** For each dialogue, we randomly sampled either a single API or a pair of connected APIs from the API graph. We also randomly sampled an action sequence to be used in the dialogue. The next key step was to generate a user query relevant to the API(s). To accomplish this, we prompted GPT-4o with the names and documentation of the API(s) and instructed it to generate a user query that covers all the API(s). For example, given two APIs "search weather station (input: coordinates, output: weather station)" and "nearby weather station coordinate API (input: location name, output: coordinates)", GPT-4o generated the query "I'm going hiking next week and would like to find a nearby weather station". This query became the first user utterance, initiating the dialogue.

**Dialogue State** The dialogue state at any point in a dialogue specifies the API name the system aims to call, its input parameters, and the parameter values provided by the user. To generate a dialogue given a user query and API(s), GPT-4o simulated concrete and plausible parameter values (e.g., "45", "264", and "NBA" in Figure 2). Dialogue states serve as a basis for generating utterances and as the ground-truth labels for dialogue state tracking (DST) evaluation (§4). The format of the dialogue state is specified in A.4.

**Scenario Instruction** Based on the dialogue states, we construct instructions to guide GPT-4o in generating user and system utterances. These instructions are based on templates.

For instance, the instruction for the dialogue in Figure 2 is as follows:

Table 2: Overall statistics of ToolDial.

| Metric | Value |
|---|---|
| Train | 8,859 |
| Validation | 1,086 |
| Test | 1,166 |
| Total | 11,111 |
| # of turns | 99,476 |
| # of turns per dialogue | 8.95 |

Table 3: Dialogue quality scores.

| Criterion | G-Eval | Humans |
|---|---|---|
| Naturalness (1–3) | 2.28 | 2.54 |
| Coherence (1–3) | 2.58 | 2.81 |
| Efficiency (1–3) | 2.81 | 2.60 |
| Faithfulness (0–1) | 0.90 | 0.95 |

- Inform intent clear: the user utters a pre-constructed query related with API LeagueHome-Standings and CategoryTournament.
- (Retriever call) → Request: the system to ask the user for **seasonId** and **tournamentId**.
- Fail inform: the user responds with seasonId **45** but fails to provide tournamentId.
- (Retriever call) → Request: the system prompts the user for **id**.
- Inform: the user responds with the requested information.
- (Call) → Request: the system asks the user for the **name** variable, to select one **id** from multiple results.
- Inform: the user responds with **NBA**.
- (Call) → Response: the system responds based on the results of the call.

By prompting GPT-4o with these scenario instructions, we create a multi-turn dialogue in which the user and system exchange utterances that align with the dialogue states to fulfill the user query (see details in A.8).

## 3.4 DIALOGUE GENERATION

**Utterance Generation** We prompt GPT-4o with simple instructions, the scenario instruction (§3.3), and the relationship between the two APIs in the API pair. Based on this guideline, GPT-4o generates each utterance of the user and the system that aligns with each turn's dialogue state (refer to the examples in Figure 2).

**Data Statistics** Our dataset ToolDial contains 11,111 dialogues in English reflecting various scenarios that can happen in the real world. The statistics of ToolDial are shown in Table 2. ToolDial is constructed based on 23 types of action sequences and has an average of 8.95 turns per dialogue.

**Data Quality** To assess the quality of our dataset, we sampled a total of 100 dialogues from all action sequences and evaluated them using both G-Eval (Liu et al., 2023) and human annotators[2]. The evaluation criteria are as follows:

- Naturalness (1–3): Are the dialogues natural interactions between the user and TALM?
- Coherence (1–3): Are the user' and the TALM's utterances relevant to and coherent with the dialogue context?
- Efficiency (1–3): Are the system's reasoning and actions to perform the user's request efficient and natural?
- Faithfulness (True or False): Are the system's responses consistent with the output of the API call?

Table 3 presents the scores from G-Eval and human annotators. On average, G-Eval assigned high scores when evaluating the 100 sample dialogues across four criteria. The dialogues received particularly high scores in Efficiency, indicating that the TALM efficiently performed the necessary steps to call APIs and collect information.

---

[2]Three Master's students majoring in data science volunteered as annotators. The authors are not included.

Table 4: Evaluation scores on three tasks. (**w GT**: ground-truth labels are included in the dialogue history, **w/o GT**: no ground-truth labels are provided)

| Model | Dialogue State Tracking | | Action Prediction | | Faithfulness |
| | w GT | w/o GT | w GT | w/o GT | w/o GT |
| --- | --- | --- | --- | --- | --- |
| GPT-3.5-turbo | 38.8 | 33.1 | 53.5 | 54.1 | 95.4 |
| GPT-4o-mini | 58.8 | 67.7 | 63.7 | 60.2 | 96.6 |
| GPT-4-turbo | 77.5 | 68.6 | 64.2 | 61.5 | **97.1** |
| GPT-4o | 81.4 | 67.8 | 57.6 | 63.7 | 96.7 |
| CodeLlama-7b-Instruct-hf | 47.2 | 28.9 | 35.7 | 30.0 | 81.7 |
| Qwen2.5-Coder-7B-Instruct | 48.9 | 34.2 | 55.8 | 46.8 | 93.9 |
| Llama3-8B-Instruct | 53.4 | 24.5 | 37.7 | 35.5 | 91.5 |
| TD-Llama | **92.7** | **72.2** | **77.5** | **91.0** | 88.4 |

**Model Biases** In ToolDial, we have leveraged several methods to mitigate GPT-4o's biases in dialogue generation. When GPT-4o generates dialogues *without any guidance*, the resulting dialogues tend to be overly repetitive and monotonous. Specifically, certain types of APIs are disproportionately preferred, and the actions performed by both the user and system lack variety, typically following a simple "Inform intent - Response" pattern. In ToolDial, we addressed this by creating dialogue data using 473 real-world APIs spanning 23 domains from RapidAPI (§3.1) and incorporating 16 actions and 23 action sequences to cover diverse scenarios (§3.2). Furthermore, for certain actions, GPT-generated utterances tend to have overly consistent speaking styles. As a solution, we predefined speaking styles for specific actions (A.7) and incorporated a mechanism to randomly select from these predefined speaking styles during the scenario instruction generation (§3.3).

## 4 EXPERIMENTS

In these experiments, we designed evaluation tasks to assess the capabilities that the TALM should possess when engaging in multi-turn interactions with users. The input to the model includes:

$$\mathcal{H}_n = (u_1, s_1, \ldots, u_n, s_n), \quad \mathcal{R}_n = (r_1, r_2, \ldots, r_n), \quad r_n = \{t_n, \mathcal{A}_n, \mathcal{RS}_n, \mathcal{D}_n, \mathcal{DS}_n\} \quad (2)$$

where $\mathcal{H}_n$ is the dialogue history up to the $n$-th turn, and $u_i$ and $s_i$ are the utterances of the user and TALM in the $i$-th turn. $\mathcal{R}_n$ represents the reasoning steps of the TALM up to the $n$-th turn, where $r_i$ is the reasoning step in turn $i$. Each reasoning step includes the thought $t$, action $\mathcal{A}$, retriever status $\mathcal{RS}$, retrieved API documentation $\mathcal{D}$ from the retriever, and dialogue state $\mathcal{DS}$ of the corresponding turn (see the formation of dialogue state and retriever status in A.4). The reasoning step of Figure 2 illustrates each component. We used $\mathcal{H}_n$ and $\mathcal{R}_n$ to predict $\mathcal{DS}$ and $\mathcal{A}$ in each turn to evaluate whether the model accurately captures the dialogue context, extracts the appropriate information, and takes the correct action. Additionally, we evaluated the last utterance $s_n$ where $\mathcal{A}_n$ ="Response" in order to assess the consistency between the model's response and the output of the API call.

### 4.1 EVALUATION TASKS

**Dialogue State Tracking** Dialogue state tracking (DST) evaluates the model's ability to determine which API should be called based on the dialogue history, as well as the accuracy of the collected input parameter values. DST can be formalized as

$$\mathcal{DS}_n = \mathcal{M}(\mathcal{H}_{n-1}, \mathcal{R}_{n-1}, u_n) \quad (3)$$

where $\mathcal{DS}_n$ is the dialogue state of turn $n$, $\mathcal{M}$ is the TALM's output, $\mathcal{H}_{n-1}$ and $\mathcal{R}_{n-1}$ are the dialogue history and the TALM's reasoning steps up to turn $n-1$. We evaluate a total of 6,746 annotated dialogue states within the test set. The evaluation checks whether the two dialogue states match completely after removing all special characters, converting to lowercase, and comparing API names, input parameters, and their corresponding values.

Table 5: F1 score for each action in the action prediction task. This indicates that fine-tuning with our data supports the system in selecting appropriate actions in multi-turn conversations.

|  |  | Response | Response fail | Request | Retriever call | Clarify | System bye | Suggest | Call |
|---|---|---|---|---|---|---|---|---|---|
| w GT | GPT-3.5-turbo | 63.8 | 0.0 | 28.4 | 66.2 | 1.3 | 95.5 | 0.0 | 53.4 |
| | GPT-4o-mini | 78.9 | 0.0 | 44.3 | 67.4 | 64.5 | 97.2 | 0.0 | 67.0 |
| | GPT-4-turbo | 93.6 | 0.0 | 18.1 | 87.5 | 56.7 | 97.2 | **29.9** | 56.4 |
| | GPT-4o | 88.3 | 0.0 | 13.7 | 74.9 | 29.6 | 97.2 | 24.6 | 54.1 |
| | Llama3-8b-Inst | 46.4 | 0.0 | 8.5 | 23.7 | 0.0 | 99.8 | 14.0 | 44.4 |
| | TD-Llama | **100.0** | **77.5** | **44.8** | **97.2** | **77.4** | **99.9** | 16.8 | **68.6** |
| w/o GT | GPT-3.5-turbo | 70.7 | 0.0 | 1.3 | 77.6 | 0.0 | 93.0 | 0.0 | 49.7 |
| | GPT-4o-mini | 88.5 | 0.0 | 36.1 | 62.6 | 0.0 | 97.2 | 0.0 | 65.1 |
| | GPT-4-turbo | 96.6 | 0.0 | 10.8 | 79.9 | 40.6 | 97.2 | 35.5 | 57.8 |
| | GPT-4o | 95.8 | 0.0 | 14.3 | 81.2 | 38.6 | 97.2 | 46.1 | 62.0 |
| | Llama3-8b-Inst | 30.5 | 0.0 | 1.9 | 27.3 | 0.0 | 93.1 | 9.4 | 42.0 |
| | TD-Llama | **98.2** | **99.1** | **78.4** | **94.5** | **99.8** | **100.0** | **99.9** | **86.9** |

**Action Prediction** The action prediction task involves selecting the next action to be taken based on the dialogue history and reasoning steps. For this task, the reasoning steps do not include ground-truth thought $t$, as it offers a direct cue for which action to take. Action prediction is formalized as

$$\mathcal{A}_n = \mathcal{M}(\mathcal{H}_{n-1}, (\mathcal{R}_{n-1} \setminus t_{n-1}), u_n) \tag{4}$$

where $\mathcal{A}_n$ is the system action in turn $n$. We evaluate a total of 9,200 annotated actions within the test set. Each turn's true action and predicted action are converted to lowercase, and special characters are removed. Evaluation is based on whether they match exactly.

**Faithfulness** We evaluate whether the final response of the TALM is grounded in the API call output, as generating responses faithful to API call results is critical for tool agents. We provide the TALM with dialogue history that includes the API call results and use G-Eval (Liu et al., 2023) to assess whether the responses reflect the API call output. The evaluation method aligns with the faithfulness criterion outlined in the Dialogue Generation step (§3.4). We evaluate a total of 943 system responses (removing "Response fail") within the test set. Following the same method as G-Eval, a GPT-4o-mini model with temperature set above 0 evaluates each response for 10 times. The average score of the 10 results (all either 0 or 1) is used as the score.

## 4.2 EXPERIMENT SETTINGS

In the real world, the model is not provided with ground-truth actions or dialogue states in the dialogue history. Hence, we evaluate models in two settings: "with GT (ground truth)" and "without GT". The latter is to see the upper bound performance of the models assuming that all prior predictions are correct. "With GT" uses the formulations in Equations 3 and 4, and "without GT" is formalized as

$$\mathcal{DS}_n^{wogt} = \mathcal{M}(\mathcal{H}_{n-1}, (\mathcal{R}_{n-1} \setminus \mathcal{DS}), u_n), \quad \mathcal{A}_n^{wogt} = \mathcal{M}(\mathcal{H}_{n-1}, (\mathcal{R}_{n-1} \setminus (t_{n-1} \cup \mathcal{A}_{n-1})), u_n) \tag{5}$$

For the faithfulness task, we only conduct the experiment in the "without GT" setting, as the model generates the final turn response and no ground-truth label exists in $\mathcal{H}_{n-1}$ or $\mathcal{R}_{n-1}$. All instruction prompts used in each task are in A.13.

As baseline models, we choose GPT-3.5-turbo, GPT-4o-mini, GPT-4-turbo, GPT-4o, CodeLlama-7b-Instruct-hf, Qwen2.5-Coder-7B-Instruct, and LLaMA3-8B-instruct. We also instruction-tuned LLaMA3-8B-instruct with the ToolDial dataset (TD-Llama) and conducted the same experiments. All experiments are conducted in a zero-shot setting, where only task-specific instructions are provided without any additional few-shot samples.

## 4.3 RESULTS

The experiment results are summarized in Table 4.

**Dialogue State Tracking** For the GPT-based models (rows 1–4), we observed that the latest versions outperform their predecessors. Additionally, both closed-source and open-source LLMs scored lower in the "w/o GT" setting compared to the "with GT" setting, as expected. Instruct-tuning the

Llama model (TD-Llama) on our dataset (row 7) significantly enhances its performance in both settings, demonstrating the value of our dataset for training TALMs. Furthermore, we observed that accuracy decreases as the number of turns increases (A.10). For TD-Llama, performance remains stable in the "with GT" setting even with longer turns. However, in the "w/o GT" setting, which better reflects real-world scenarios, performance declines as the number of turns increases. This suggests that dialogue state tracking over multiple turns in real-world settings remains a challenging task. A detailed error analysis of DST is provided in A.9.

**Action Prediction**    In the action prediction task, GPT models (rows 1–4) achieved an accuracy of around 60%, which suggests that there is significant room for improvement. On the other hand, Llama3-8B-Instruct received a much lower accuracy of around 35%, indicating the difficulty in determining appropriate actions based on dialogue history. However, once fine-tuned on our dataset, TD-Llama (row 7) achieved an accuracy of 77.5% and 91.0% on with GT and w/o GT respectively, outperforming GPT models.

To better understand the models' performance across actions, Table 5 shows the F1-score for each action. Here, GPT models show relatively low scores for predicting actions like "Request", "Clarify", and "Suggest". This result is consistent with our observation that GPT-based models often rush to provide answers without collecting further information or asking clarifying questions. These actions are essential in real-world interactions to serve the user's needs precisely and reduce hallucinations, and TD-Llama demonstrates improved performance on these actions. Another notable result is the low performance of GPT models on the "Response fail" action. When the user refuses to proceed with a suggested API, the models often attempt to clarify the user's intent ("Clarify") rather than acknowledging the failure and terminating the dialogue. While this move could be considered somewhat reasonable, it violates the instruction provided in the prompt and may bother the user.

**Faithfulness**    GPT models achieved over a 90% accuracy in the faithfulness task. However, the performance of the smaller Llama-based models remains around 88.4%. This demonstrates that small language models are vulnerable to hallucination, and we need better methods for improving the faithfulness of these models.

**Overall Performance**    To accurately resolve a user's query in real-world settings, generating correct reasoning trace (dialogue state, action) based on dialogue history and the user's most recent utterance is crucial at each turn. We evaluated the overall performance of the fine-tuned TD-Llama model in this context. We assessed whether the model correctly generated both the dialogue state and action after processing 5,213 user utterances in the test set. A result was marked as true if both the action and dialogue state were accurately generated for each reasoning step; otherwise, it was marked as false. This evaluation yielded a performance score of 77.1%. Additionally, for 1,166 test dialogues, we measured the proportion of dialogues in which the reasoning trace was correctly generated for all turns—from the first to the last— achieving an accuracy rate of approximately 28.3%. This suggests that there is significant room for improvement in overall performance.

## 5    CONCLUSION

In this work, we introduce ToolDial, a multi-turn dialogue dataset that reflects interactions between a user and the TALM in real-world scenarios. To generate realistic dialogues, we construct and employ an API graph representing the interdependencies between APIs, aiming to simulate scenarios in which the TALM must call multiple APIs to obtain necessary information. Additionally, we define 16 user and system actions to reflect the rich dynamics of tool-use conversations. To generate a dialogue, we first sample APIs and an action sequence as a skeleton. This skeleton is then augmented with dialogue states specific to the APIs and finally converted into utterances using GPT-4o. Our evaluation demonstrates that modern language models perform poorly in predicting appropriate actions and dialogue states in complex multi-turn interactions. We believe ToolDial can serve as a valuable resource for advancing the field of TALM.

ACKNOWLEDGEMENTS

This work was supported by the New Faculty Startup Fund and the Creative-Pioneering Researchers Program through Seoul National University. It was also supported by the National Research Foundation of Korea (NRF) grants (RS-2024-00333484, RS-2024-00414981) and the Institute of Information & communications Technology Planning & Evaluation (IITP) under the Leading Generative AI Human Resources Development (IITP-2025-RS-2024-00397085) grant, both funded by the Korea government (MSIT, Ministry of Science and ICT).

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

# A APPENDIX

## A.1 ENTITY DESCRIPTION GENERATION

Table 6: Prompt used to generate input entity

| Prompt for generating input entity descriptions |
| --- |
| **System** |
| You are an intelligent annotator. Your mission is to write the description of input parameters more specifically, referring to the given information. |
| Write as specifically as possible, referring to the given information. The new description should be based on the existing description but rewritten to better reflect the content of the API description and API endpoint description than before. Just return the input and its description, not individual words. For example: |
| Category of the API: Data Description of the Category: APIs facilitate the seamless exchange of data between applications and databases, enabling developers to integrate functionalities securely and swiftly. API Name: YouTube Media Downloader API Description: A scraper API for YouTube search and download. Get videos, subtitles, comments without age or region limits (proxy URL supported). API Endpoint Name: Get Channel Details API Endpoint Description: This endpoint fetches details of a YouTube channel. |
| List of input parameters: |
| Input parameter name: channelId Description: Channel ID, custom URL name, or handle. @ is required as a prefix for a channel handle. |
| Input parameter name: lang Description: Language code (ISO-639) for localized results. Defaults to en-US. Unsupported codes will fallback to en-US. |
| For this, you should return: |
| [ ["channelId", "The unique identifier for the YouTube channel, which can be the channel's ID, a custom URL name, or a channel handle. When using a channel handle, ensure to prefix it with '@' (e.g., '@channelname')".], ["lang", "The language code (ISO-639) used to specify the language for the localized results. If not provided, the default is 'en-US'. In case an unsupported language code is supplied, the results will revert to 'en-US'".] ] Now, I'll give you another description. Follow the instructions, referring to the example. |
| Write as specifically as possible, referring to the given information. The new description should be based on the existing description but written in a way that better reflects the content of the API description and API endpoint description than before. Just return the input and its description, not individual words. |

Table 7: Prompt used to generate output entity description

| Prompt for generating output entity descriptions |
| --- |
| **System** |
| You are an intelligent annotator. Your mission is to write the description of the output components of an API endpoint, referring to the given information below. |
| For example: |
| Category of the API: Data Description of the Category: APIs facilitate the seamless exchange of data between applications and databases, enabling developers to integrate functionalities securely and swiftly. API Name: YouTube Media Downloader API Description: A scraper API for YouTube search and download. Get videos, subtitles, comments without age or region limits (proxy URL supported). API Endpoint Name: Get Channel Details API Endpoint Description: This endpoint fetches details of a YouTube channel. |
| Based on the given description, write the description of the output component of this API endpoint. Write as specifically as possible. Do not generate examples for each component. The description should reflect as closely as possible the description of the API and the API endpoint, so that even someone seeing this API endpoint for the first time can understand exactly what the output component means. (A component separated with — refers to the hierarchy of the schema. For example, avatar—height refers to the height of the avatar.) |

Output components: [ { 'name': status" }, { 'name': type" }, { 'name': id" }, { 'name': name" }, { 'name': handle" }, { 'name': description" }, { 'name': isVerified" }, { 'name': isVerifiedArtist" }, { 'name': subscriberCountText" }, { 'name': videoCountText" }, { 'name': viewCountText" }, { 'name': joinedDateText" }, { 'name': country" }, { 'name': links—title" }, { 'name': links—url" }, { 'name': avatar—url" }, { 'name': avatar—width" }, { 'name': avatar—height" } ]

For this example, you have to return,

[ { 'name': status", 'description": Indicates whether the API call was successful. True means the call was successful, while False means it failed" }, { 'name': type", 'description": Specifies the type of YouTube channel, such as 'User' or 'Brand', indicating the category of the channel". }, { 'name': id", 'description": The unique identifier assigned to the YouTube channel, which can be used to reference the channel in other API calls or services". }, { 'name': name", 'description": The official name of the YouTube channel as displayed on the platform, which is set by the channel owner". }, { 'name': handle", 'description": The unique handle of the YouTube channel, which often appears in the URL of the channel's page". }, { 'name': description", 'description": A textual description provided by the channel owner that gives an overview of the channel's content, themes, and purpose". }, { 'name': isVerified", 'description": Indicates whether the YouTube channel is verified by YouTube. A verified status signifies authenticity and is usually granted to public figures, brands, and popular content creators". }, { 'name': isVerifiedArtist", 'description": Specifies if the YouTube channel is recognized as a verified artist's channel, which is a special status for musicians and bands to highlight their official content". }, { 'name': subscriberCountText", 'description": A human-readable representation of the number of subscribers the channel has, formatted for display purposes". }, { 'name': videoCountText", 'description": A human-readable representation of the total number of videos uploaded by the channel, formatted for display purposes". }, { 'name': view-CountText", 'description": A human-readable representation of the total number of views across all videos on the channel, formatted for display purposes". }, { 'name': joinedDateText", 'description": A human-readable representation of the date when the YouTube channel was created, formatted for display purposes". }, { 'name': country", 'description": The country where the YouTube channel is registered or primarily based, providing geographical context". }, { 'name': links—title", 'description": The title of an external link provided by the channel, which can lead to the channel's social media profiles, websites, or other related content". }, { 'name': links—url", 'description": The URL of an external link associated with the channel, which directs users to other online presences of the channel". }, { 'name': avatar—url", 'description": The URL of the channel's avatar image, which is the profile picture displayed on the channel's page". }, { 'name': avatar—width", 'description": The width of the avatar image in pixels, providing information about the image dimensions". }, { 'name': avatar—height", 'description": The height of the avatar image in pixels, providing information about the image dimensions". } ]

Now, I'll give you another API endpoint description. Write the description of the output components and return it in the same format as the example. Just return the result, not individual words. Based on the given description, write the description of the output components of this API endpoint. Write as specifically as possible. Do not generate examples for each component. The description should reflect the API and the API endpoint as closely as possible, so that even someone seeing this API endpoint for the first time can understand exactly what the output component means. (A component separated with — refers to the hierarchy of the schema. For example, avatar—height refers to the height of the avatar.)

Fill the <Your response>.

<Your response>

## A.2 KEYWORDS EXTRACTION

Table 8: Prompt used to extract keywords

| Prompt for extracting keywords |
|---|
| **System** |
| Extract the keywords from the given paragraph. Prioritize proper nouns first and nouns second, selecting up to 4 words that best describe the paragraph. Return the keywords in CSV format. Remember, the maximum is 4 words. |
| Paragraph: |

## A.3 USER AND SYSTEM ACTION LIST

Our work defines 8 user actions and 8 system actions, which form the basis for conceptualizing interactions. Table 9 and 10 provide the names and descriptions of these actions.

Table 9: User action and description

| User Action | Description |
|---|---|
| Inform intent clear | Say what one wants specifically. |
| Inform intent clear add | Say what one wants specifically with the information of input parameter. |
| Inform intent vague | Say what one wants vaguely. |
| Inform | Inform the requested information to system. |
| Fail inform | Fail to reply to system's request. |
| Affirm | Agree to the system's proposition. |
| Negate | Deny the system's proposal. |
| User bye | Say thank you and goodbye to system. |

Table 10: TALM action and description

| System Action | Description |
|---|---|
| Request | Asks some information to user. |
| Response | Reply to user's request based on the result of API call. |
| Clarify | If user's query is vague, re-ask user to get intent specifically. |
| Suggest | Making a suggestion for an unclear user's intent and asking whether it satisfies the user. |
| Response fail | Notify the user that the system cannot execute the request due to insufficient information. |
| System bye | System says goodbye to user politely. |
| Call | Call the API with collected information from user or else and don't reply to user yet. |
| Retriever call | Call the retriever to find proper API to satisfy user's requests. |

## A.4 DIALOGUE STATE AND RETRIEVER STATUS ANNOTATION FORMAT

Our data is annotated with "retriever status" each turn. This indicates whether the retriever was called for each turn of the conversation, the APIs retrieved as a result, and their respective retriever scores. The actions that the TALM should take vary depending on the retriever score. If there is an API with a score of 0.6 or higher, the TALM asks the user for input parameters to call it. If the score is between 0.5 and 0.6, the TALM suggests the retrieved API, and if the score is lower, it asks for clarification of the user's query. Format of retriever status can have three types described below:

- **When retriever is not called**
  {Retriever status: false, Retrieved API: none}

- **Situation where the TALM needs to find the appropriate API to solve the user's query.**
  {Retriever status: true, Retrieved API: {API 1: 0.65, API2: 0.54, API3: 0.51...}}

- **Situation that TALM needs to obtain an input parameter that the user has not provided.**
  {Retriever status: true, Retrieved API: [Output component of source API to procure target API's input parameter param1 → output1]}

Additionally, our dataset is labeled with the dialogue state for each turn. The dialogue state includes the API that the TALM is currently attempting to execute and the input parameter information collected for that API, based on the dialogue history. The dialogue state has the following format:

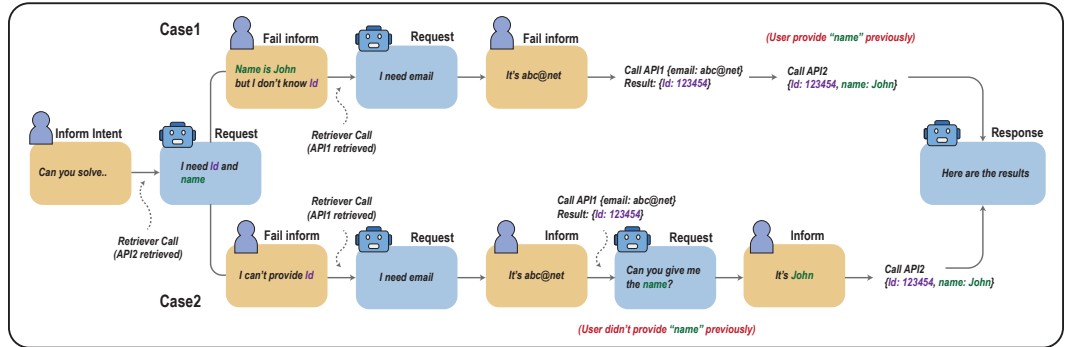

Figure 4: Possible cases of two action sequences according to perform types "Fail inform".

- **When there is no confirmed API**
  {API confirmed: false, API status: none}

- **When the API is confirmed**
  {API confirmed: true, API status: {API name: "API1", Required parameters: {param1: " ", param2: " "}, Optional parameters: {param3: " "}}}

- **When the API is confirmed and some input parameter information can be extracted from dialogue history**
  {API confirmed: true, API status: {API name: "API1", Required parameters: {param1: "value1", param2: " "}, Optional parameters: {param3: "value3"}}}

## A.5 VARIATION OF FAIL INFORM ACTION

User can perform "Fail inform" in two ways: either indicating they don't know one parameter while providing the rest, or simply stating they don't know the missing parameter without further input. Figure 4 demonstrates the two ways.

## A.6 COMPREHENSIVE ACTION SEQUENCES

Assuming that at most two APIs are called in a dialogue, a total of 23 action sequences are derived for data generation. Among these, 15 sequences involve two APIs, 7 involve one API, and 1 involves a failure to call any APIs. The 15 sequences with two APIs are further categorized based on the type of action sequence request: either directly requesting input parameters from the user ("Request") or making an additional requesting to select an appropriate value from multiple results ("Request-multi").

Table 11: Action Sequences with two APIs

| No. | Action Sequence |
| --- | --- |
| 1 | 'Inform intent vague', 'Retriever call', 'Suggest', 'Affirm', 'Request', 'Fail inform', 'Retriever call', 'Request', 'Inform', 'Call', 'Call', 'Response', 'User bye', 'System bye' |
| 2 | 'Inform intent vague', 'Retriever call', 'Suggest', 'Affirm', 'Request', 'Fail inform', 'Retriever call', 'Call', 'Request', 'Inform', 'Call', 'Response', 'User bye', 'System bye' |
| 3 | 'Inform intent vague', 'Retriever call', 'Clarify', 'Inform intent clear', 'Retriever call', 'Request', 'Fail inform', 'Retriever call', 'Call', 'Request', 'Inform', 'Call', 'Response', 'User bye', 'System bye' |
| 4 | 'Inform intent vague', 'Retriever call', 'Clarify', 'Inform intent clear', 'Retriever call', 'Request', 'Fail inform', 'Retriever call', 'Request', 'Inform', 'Call', 'Request-multi', 'Inform', 'Call', 'Response', 'User bye', 'System bye' |
| 5 | 'Inform intent vague', 'Retriever call', 'Clarify', 'Inform intent clear', 'Retriever call', 'Request', 'Fail inform', 'Retriever call', 'Request', 'Inform', 'Call', 'Request', 'Inform', 'Call', 'Response', 'User bye', 'System bye' |

| | |
|---|---|
| 6 | 'Inform intent clear', 'Retriever call', 'Request', 'Fail inform', 'Retriever call', 'Call', 'Request-multi', 'Inform', 'Call', 'Response', 'User bye', 'System bye' |
| 7 | 'Inform intent clear', 'Retriever call', 'Request', 'Fail inform', 'Retriever call', 'Request', 'Inform', 'Call', 'Request', 'Inform', 'Call', 'Response', 'User bye', 'System bye' |
| 8 | 'Inform intent clear', 'Retriever call', 'Request', 'Fail inform', 'Retriever call', 'Request', 'Inform', 'Call', 'Call', 'Response', 'User bye', 'System bye' |
| 9 | 'Inform intent clear', 'Retriever call', 'Request', 'Fail inform', 'Retriever call', 'Call', 'Request', 'Inform', 'Call', 'Response', 'User bye', 'System bye' |
| 10 | 'Inform intent vague', 'Retriever call', 'Suggest', 'Affirm', 'Request', 'Fail inform', 'Retriever_call', 'Request', 'Inform', 'Call', 'Request-multi', 'Inform', 'Call', 'Response', 'User bye', 'System bye' |
| 11 | 'Inform intent vague', 'Retriever call', 'Clarify', 'Inform intent clear', 'Retriever call', 'Request', 'Fail inform', 'Retriever call', 'Request', 'Inform', 'Call', 'Call', 'Response', 'User bye', 'System bye' |
| 12 | 'Inform intent vague', 'Retriever call', 'Suggest', 'Affirm', 'Request', 'Fail inform', 'Retriever call', 'Request', 'Inform', 'Call', 'Request', 'Inform', 'Call', 'Response', 'User bye', 'System bye' |
| 13 | 'Inform intent clear', 'Retriever call', 'Request', 'Fail inform', 'Retriever call', 'Request', 'Inform', 'Call', 'Request-multi', 'Inform', 'Call', 'Response', 'User bye', 'System bye' |
| 14 | 'Inform intent vague', 'Retriever call', 'Clarify', 'Inform intent clear', 'Retriever call', 'Request', 'Fail inform', 'Retriever call', 'Call', 'Request-multi', 'Inform', 'Call', 'Response', 'User bye', 'System bye' |
| 15 | 'Inform intent vague', 'Retriever call', 'Suggest', 'Affirm', 'Request', 'Fail inform', 'Retriever call', 'Call', 'Request-multi', 'Inform', 'Call', 'Response', 'User bye', 'System bye' |

Table 12: Action Sequences with one API

| No. | Action Sequence |
|---|---|
| 1 | 'Inform intent vague', 'Retriever call', 'Clarify', 'Inform intent clear', 'Retriever call', 'Request', 'Inform', 'Call', 'Response', 'User bye', 'System bye' |
| 2 | 'Inform intent vague', 'Retriever call', 'Clarify', 'Inform intent clear add', 'Retriever call', 'Call', 'Response', 'User bye', 'System bye' |
| 3 | 'Inform intent vague', 'Retriever call', 'Suggest', 'Affirm', 'Request', 'Inform', 'Call', 'Response', 'User bye', 'System bye' |
| 4 | 'Inform intent vague', 'Retriever call', 'Clarify', 'Inform intent clear add', 'Retriever call', 'Request', 'Inform', 'Call', 'Response', 'User bye', 'System bye' |
| 5 | 'Inform intent clear add', 'Retriever call', 'Request', 'Inform', 'Call', 'Response', 'User bye', 'System bye' |
| 6 | 'Inform intent clear add', 'Retriever call', 'Call', 'Response', 'User bye', 'System bye' |
| 7 | 'Inform intent clear', 'Retriever call', 'Request', 'Inform', 'Call', 'Response', 'User bye', 'System bye' |

Table 13: Action Sequence with failure

| No. | Action Sequence |
|---|---|
| 1 | 'Inform intent vague', 'Retriever call', 'Suggest', 'Negate', 'Response fail' |

## A.7 UTTERANCE STYLE

We have defined several utterance styles for some actions to prevent GPT-4o from generating consistent speaking styles.

- User Action
  - Inform
    * Sure! ∼, Ok ∼, Certainly!
  - Affirm
    * Yes, that works., That would be great., Sure, that sounds good., Yes, please proceed.
  - Negate
    * No, that's not what I meant, I'm good. Thank you though, Umm... that's not what I want
- System Action
  - Request
    * To call ∼, I need ∼, May I ask for ∼, Please tell me ∼,
  - Clarify
    * Could you please provide more ∼, I'm not sure I understand. Can you clarify ∼, Could you explain that in more ∼, Can you clarify your ∼

## A.8   SCENARIO INSTRUCTION

We use detailed dialogue scenario instruction to ensure that the predefined interactions are accurately reflected in the dialogue data and that the correct entities are included in each utterance.

Table 14: Example of scenario instruction

| Scenario prompt |
|---|

**User turn**
-user action: Inform intent vague (Say what one wants vaguely.)
-situation: User requests something from the system. User says "Can you provide detailed information about a city I plan to visit, including its geographical context and population data, so I can find some highly-rated local businesses with good reviews and contact details nearby?"

**System turn**
-system action: Retriever call (Call the retriever to find the proper API to satisfy the user's request.)
-situation: The system, having received the user's query, calls the retriever to find an appropriate API. In this turn, the system's thought is, "The user seems to have intent. I will call the retriever".
Retriever status: retriever_call: 'true', retrieved_api: 'Data—local_business_data—Search Nearby': 0.56445915, 'Data—local_business_data—Search In Area': 0.5539355, 'Mapping—places—Place properties': 0.5367253, 'Location—spott—Search places': 0.53351307, 'Data—serpwow—Google Place and Maps Details': 0.5169816
Dialogue state: api_confirmed: 'false', api_status: 'none'

**System turn**
-system action: Suggest (Make a suggestion for an unclear user intent and ask whether it satisfies the user.)
-situation: Since the user's query is unclear, no API with a retriever score higher than 0.6 has been found. However, several APIs have scores between 0.5 and 0.6. The system asks whether it would be appropriate to run Data—local_business_data—Search Nearby, which has the highest score among them, and retrieve the result. At this time, the system does not mention the name of the API directly.
Retriever status: retriever_call: 'false', retrieved_api: 'none'
Dialogue state: api_confirmed: 'false', api_status: 'none'

**User turn**
-user action: Affirm (Agree to the system's proposition.)

-situation: User agrees with the system's proposition. User's speech should follow this format: "Yes, please proceed".

---

**System turn**
-system action: Request (Asks some information to user.)
-situation: System asks user to..

---

## A.9 DST ERROR ANALYSIS

Table 15: DST error analysis for GPT-based models

|  | GPT-3.5-turbo | | GPT-4o-mini | | GPT-4-turbo | | GPT-4o | |
|---|---|---|---|---|---|---|---|---|
|  | W GT | W/O GT | W GT | W/O GT | W GT | W/O GT | W GT | W/O GT |
| # of Error | 4128 | 4512 | 2781 | 2177 | 1515 | 2117 | 1257 | 2169 |
| Generation Err | 0 | 0 | 0 | 0 | 0 | 0 | 0 | 0 |
| API Conf Err (GT = T) | 1609 | 1841 | 1060 | 504 | 211 | 224 | 243 | 1607 |
| API Conf Err (GT = F) | 750 | 410 | 848 | 373 | 692 | 891 | 343 | 133 |
| Format Err | 532 | 502 | 153 | 0 | 0 | 74 | 0 | 531 |
| Slot Err | 1139 | 1674 | 508 | 912 | 430 | 774 | 443 | 221 |
| Value Err | 561 | 630 | 398 | 823 | 498 | 626 | 495 | 221 |

Table 16: DST error analysis for Llama3-8b-instruct and TD-llama

|  | Llama3-8b-instruct | | TD-llama | |
|---|---|---|---|---|
|  | W GT | W/O GT | W GT | W/O GT |
| # of Err | 3138 | 5090 | 492 | 1873 |
| Generation Err | 3 | 0 | 260 | 1619 |
| API Conf Err (GT = T) | 583 | 1014 | 30 | 1 |
| API Conf Err (GT = F) | 723 | 923 | 0 | 0 |
| Format Err | 531 | 319 | 61 | 103 |
| Slot Err | 1101 | 2663 | 6 | 23 |
| Value Err | 846 | 1423 | 134 | 144 |

Tables 15 and Table 16 present the error analysis results for each model on the dialogue state tracking (DST) task. We categorized the errors in DST as follows.

- **Generation Error:** This occurs when the dialogue state dictionary is not generated at all.
- **API Confirmation Error (GT = True):** This error happens when the API is confirmed (`api_confirmed=true`), but is incorrectly predicted as not confirmed (`api_confirmed=false`).
- **API Confirmation Error (GT = False):** This error occurs when the API is not confirmed (`api_confirmed=false`), but the model incorrectly predicts it as confirmed (`api_confirmed=true`).
- **Format Error:** This occurs when the dialogue state does not fully generate all fields such as `api_confirmed`, `api_status`, required parameters, and optional parameters.
- **Slot Error:** When `api_confirmed` is true, this error involves generating a dialogue state that does not include all required and optional parameter slots as specified in the API documentation.
- **Value Error:** This error involves incorrectly extracting the slot's value from the dialogue history, with the following types:
  - **Extracting Input Value from Multiple Result Error:** This error occurs when an appropriate value cannot be selected from multiple results returned by the API output (as seen in turns 6 and 7 of Figure 2).

- **Inform Intent Add Error:** This occurs when there is a value within the user query that could be used as an input parameter (Inform intent clear add), but the model fails to track it.
- **Other General Input Parameter Extraction Errors:** Errors that occur in typical situations where the input parameter is extracted incorrectly.

The error analysis tables show error counts, where slot and value errors can overlap in a single prediction, causing their sum to exceed the total errors. We also provide examples of DST errors as part of a qualitative error analysis.

------------------------------------------------------------------

**<Example1. Extracting Input Value from Multiple Result Error>**

...
- Retriever status:{'Retriever call':'true', 'retrieved_api': ['getPaymentInitiationInstructionSummary','Output to procure input parameter uetr of getPaymentInitiationInstruction: end_to_end_identification']} ...
- Call result:

```
[{'end_to_end_identification': 'XYZ/123-45678/2021-07-15',
  'creation_date_time': '2022-05-20T14:30:00',
  'requested_execution_date': '2021-10-01T09:00:00',
  'instructed_amount': '1500000.50'},
 {'end_to_end_identification': 'XYZ/123-45679/2021-07-16',
  'creation_date_time': '2022-05-25T10:15:00',
  'requested_execution_date': '2021-10-02T09:00:00',
  'instructed_amount': '750000.00'}]
```

- Thought: The API call to 'getPaymentInitiationInstructionSummary' returned multiple results. I need to ask the user to select one based on the **'requested execution date'**. ...
- Message: Please tell me the requested execution date of the transaction you are interested in: **'2021-10-01T09:00:00' or '2021-10-02T09:00:00'**?

------------------------------------------------------------------

**Label:** {...{'api_name':'getPaymentInitiationInstruction','required_parameters': {**'uetr':''**},...}}
**Predict:**{...{'api_name':'getPaymentInitiationInstruction','required_parameters':{**'uetr':'XYZ/123-45678/2021-07-15'**},...}}

------------------------------------------------------------------

Example 1 is a situation where the output 'end to end identification' of API 'getPaymentInitiationInstructionSummary' can procure the 'uter' of 'getPaymentInitiationInstruction'. In this turn, multiple 'end to end identification' values are returned, requiring a request to the user to select one value and gather the uter value accordingly. However, it was observed that the model's prediction arbitrarily selected one of the results, which leads to generate wrong dialogue state.

------------------------------------------------------------------

**<Example2. Inform Intent Add Error Error>**

User: How do I create a **Basic Plan** for recurring billing payments?
System:
...
(retrieved createPlan from the retriever)
...
- API docs: {'api_name':'createPlan',
{'input_parameter_name': 'name',
'description': 'The name of the billing plan that is being created for the purpose of managing payment schedules and billing cycles in the PayPal payment processing system.',
...(and other input parameter's name and descriptions)...
},
- Message: To call the API to create a Basic Plan, I need the following information: accessToken, description, paymentDefinitions, type, merchantPreferences, and sandbox.

------------------------------------------------------------------

**Label:** {...{'api_name': 'createPlan', 'required_parameters': {'accessToken': '', 'description':

'', 'paymentDefinitions': '', **'name': 'Basic Plan'**, 'type': '', 'merchantPreferences': ''}, 'optional_parameters': {'sandbox': ''}}}

**Predict:** {...{'api_name': 'createPlan', 'required_parameters': {'accessToken': '', 'description': '', 'paymentDefinitions': '', **'name': ''**, 'type': '', 'merchantPreferences': ''}, 'optional_parameters': {'sandbox': ''}}}

――――――――――――――――――――――――――――――――――――――――――――――――――――――

Example 2 is a case where the input parameter 'name' required for executing the 'createPlan' API is specified as the value 'Basic Plan' in user's query. Additionally, the system's request action message only inquires about input parameters other than 'name'. In such a situation, the dialogue state should be generated with 'name' already populated as 'Basic Plan'. However, it was generated with 'name' left empty, resulting in this case being classified as an error.

――――――――――――――――――――――――――――――――――――――――――――――――――――――

## A.10   DST ACCURACY BASED ON TURN LENGTH

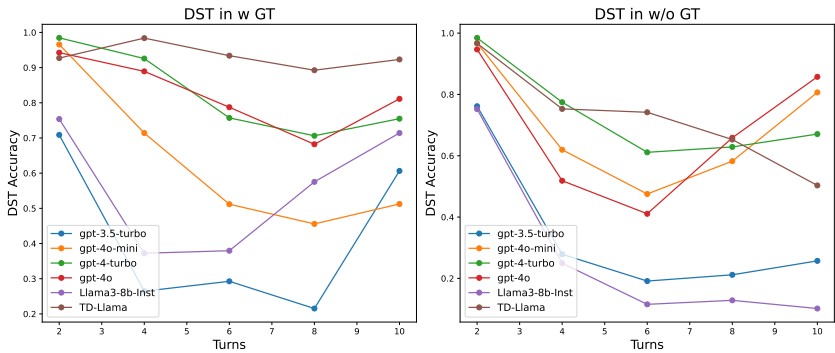

Figure 5: DST Accuracy for each model as the number of dialogue turns increases.

## A.11   REMOVING MISMATCH ERRORS

Blow examples shows the mismatch errors that occur during edge construction. There is a domain mismatch and an entity mismatch.

**Domain mismatch**

**API 1**

- **Domain and Tools: Sports basketapi**
- API name: LeagueTopPlayersPlayoffs
- Entity name: tournamentId
- Entity Description: The id of the specific basketball tournament for which the top players in the playoffs are being retrieved.

**API 2**

- **Domain and Tools: Sports baseballapi**
- API name: PlayerRegularSeasonStatistics
- Entity name: tournamentId
- Entity Description: The id of the specific baseball tournament for which the regular season statistics of a player are being requested.

**Entity mismatch**

**API 1**

- Domain and Tools: Sports icehockeyapi
- API name: PlayerRegularSeasonStatistics
- **Entity name: playerId**
- Entity Description: The unique identifier for a specific ice hockey player whose regular season statistics are being requested.

**API 2**

- Domain and Tools: Sports icehockeyapi
- API name: LeaguePlayoffsTopPlayers
- **Entity name: seasonId**
- Entity Description: The id of the specific ice hockey season for which the top players are being retrieved during the playoffs.

## A.12 PROMPT FORMAT FOR THE EXPERIMENT

Table 18 presents the prompt format used in the experiments conducted in our work. Both open-source and closed-source LLMs utilized this format. DST involves predicting all dialogue states present in the format for each dialogue, while action prediction focuses on predicting all actions. In the case of action prediction, all "thought" within the format are removed prior to the task. The W/O GT setting requires predicting the dialogue state and action for each turn using the dialogue history in the format without any dialogue states or actions included in the reasoning steps (for DST and action prediction, respectively).

## A.13 EVALUATION PROMPTS

We release all the prompts used in our experiments. Table 17 contains the prompt used for evaluating edges in graph construction (§3.1), Table 19 includes the prompt used for dialogue state tracking evaluation, Table 20 provides the prompt used for action prediction evaluation, and Table 21 presents the prompt used for faithfulness evaluation. The prompt used in the overall performance task is detailed in the provided link[3].

---

[3]https://github.com/holi-lab/ToolDial/blob/main/experiments/prompts.py

Table 17: Prompt used to evaluate edges.

| Edge Evaluation Prompt |
|---|

**System**

Your task is to determine whether the source attribute in the response from the source API is compatible with the api input of the target API. Then, craft a JSON formatted response that aligns with the expected output of the API, guided by the provided examples. For your judgment, we will provide descriptions of tool description, API Documentation, source attribute and target attribute of both APIs.

The judgment is a two step process. In the first step, determine whether the two attributes are compatible based on a deep understanding of the source attribute and target attribute. Determine whether the source attribute and target attribute are compatible through attribute descriptions. The second step is to determine whether the input of the target API is compatible with the intent of the target API. If both steps are considered compatible, follow the Output format for True to output the result. If not, follow the Output format for False to output the result. Your responses must adhere to a specific JSON structure, which is as follows:

Output format for True:

```
{"error": "","response": "<Your_Response>"}
```

Output format for False:

```
{"error": "Invalid Edge Error","response": "<Your_Response>"}
```

The response field should contain the content you formulate based on the API's functionality and the input provided. Ensure that your responses are meaningful, directly addressing the API's intended functionality. If the provided examples are mostly error messages or lack substantial content, use your judgment to create relevant and accurate responses. The key is to maintain the JSON format's integrity while ensuring that your response is an accurate reflection of the API's intended output within the tool. Please note that your answer should not contain anything other than a json format object, which should be parsable directly to json.

**Note that:**

- Your response should be around 100 to 200 words, containing rich information given the api input parameters. Keep Your answer short and simple.

- Your response must be effective and have practical content.

- If the api response example if null or ineffective, ignore the example and give your independent response.

**User**

**API Documentation:**

source API Documentation JSON file
target API Documentation JSON file
source attribute: description of source attribute
tource attribute: description of target attribute

**API Examples:**

Example input 1: Example response 1
Example input 2: Example response 2
Example input 3: Example response 3

**API Input:**

Argument JSON string, e.g:

```
{"category":"Logistics", "tool_name": "SQUAKE",
"api_name": "Checkhealth", "tool_input": "{}",
"strip": "filter"}
```

Table 18: Prompt Format for the Experiment

| **Prompt format used in our experiments** |
| --- |
| **User:** Can you provide insights into which articles on NYTimes.com have been shared the most on social media platforms, particularly Facebook, to understand active engagement trends?

**System:**
- Thought: The user seems to have intent. I will call the retriever.
- Action: retriever call
- Retriever status: {retriever_call: true, retrieved_api: {NYTimesAPI: 0.71, … }}
- API documentation: {api_name: NYTimesAPI, (with required and optional parameter informations)}
- Thought: Based on results of the retriever, I need to call NYTimesAPI. To call it, I need to ask "share_type" and "period" to the user.
- Action: request
- Retriever status: {retriever_call: false, retrieved_api: none}
- Message: Could you please provide the type of sharing on Facebook for articles and the time period for retrieving popular articles?
- Dialogue State: <Current dialogue state>

**User:** Certainly! The share type is Facebook and the period is 7 days.

**System:**
- Dialogue State: <Current dialogue state>
- Thought: Based on the user's response, now I can call the API NYTimesAPI
- Action: call … |

Table 19: Instruction prompt for the Dialogue State Tracking task

---

**Dialogue state tracking task evaluation prompt**

**System**

Instruction: You will be given part of a dialogue between the user and the system. In this dialogue, the user is requesting information from the system, and the system will execute an API call to retrieve the necessary information.

Your task is to output the appropriate dialogue state for the current turn, based on the dialogue provided.

System Rules:

1. The system selects the API with the highest score from among the APIs in the retriever status that have a score of 0.6 or higher and are suitable for processing the user's query.

2. If no API has a score higher than 0.6, the system cannot confirm the API to call.

Dialogue state format:

Case 1. When the API has not been confirmed (if the retrieved API does not have a score of 0.6 or higher):

```
{'api_confirmed': 'false', 'api_status': 'none'}
```

- The API is not confirmed, so api_confirmed is set to false.

- Therefore, api_status is 'none'.

- If api_confirmed is false, api_status must be 'none'.

Case 2. When the API is confirmed (if the retrieved API has a score of 0.6 or higher):

```
{'api_confirmed': 'true', 'api_status': {'api_name': 'API1',
'required_parameters': {'param1': '', 'param2': 'value1'},
'optional_parameters': {'param3': ''}}}
```

- The API is confirmed, so api_confirmed is set to true.

- The api_status contains the name of the API and the input parameter list needed for the API call. Any parameter whose value can be extracted from the dialogue history will have its value filled in.

- The 'param1', 'param2', and 'param3' in Case 2 are just example values. Do not use these parameters. Refer to the given API documentation on each turn.

- The input parameters should always be determined by consulting the API documentation. Do not hallucinate them.

Now, part of the dialogue will be given. Just generate the dialogue state in the given format, without adding any extra words.

Dialogue:

```
{dialogue_history}
```

Table 20: Instruction prompt for the Action prediction task

| **Action prediction task evaluation prompt** |
| --- |
| **System** |

**System**

Instruction: You will be given part of a dialogue between the user and the system. In this dialogue, the user is requesting information from the system, and the system will execute an API call to retrieve the necessary information.

Your task is to predict the action that the system should take after the last utterance of the user. Read the dialogue history and return the one action that is most appropriate for the system to take next. The actions that the system can take are as follows:

- Request: Asks the user for some information.
- Response: Replies to the user's request based on the result of the API call.
- Clarify: If the user's query is vague, re-ask the user to specify their intent. If there is no API in the most recently retrieved results with a score above 0.5, "Clarify" is required.
- Suggest: Makes a suggestion for an unclear user's intent and asks whether it satisfies the user. If there is an API in the most recently retrieved results with a score above 0.5 but none exceeding 0.6, a "Suggest" action is required.
- Response fail: Notifies the user that the system cannot execute the request due to insufficient information.
- System bye: Politely says goodbye to the user.
- Call: Calls the API with the collected information from the user or other sources but does not reply to the user yet.
- Retriever call: Calls the retriever to find the proper API to satisfy the user's request. The system should call the retriever in the following two situations:
    1. When the user specifies a requirement, and the system needs to search for an API to fulfill it.
    2. When the user does not provide the input parameters required for an API call, and the system needs to search for another API to obtain those parameters.

Of the eight actions given, return only the one that you think is most appropriate. Do not return any value other than the action provided above. Just return the action, not a single word more.
Dialogue History:

```
{dialogue_history}
```

Table 21: Instruction prompt for the Faithfulness task

| **Faithfulness task evaluation prompt** |
| --- |
| **System** |

**System**

Instruction: You will be given part of a dialogue between the user and the system. In this dialogue, the user is requesting information from the system, and the system will execute an API call to retrieve the necessary information. Your task is to generate a response that satisfies the user's initial query based on the API call results provided in the dialogue history.
Dialogue History:

```
{dialogue_history}
```

