# OpenReview forum: "ToolDial: Multi-turn Dialogue Generation Method for Tool-Augmented Language Models"
_ICLR.cc/2025/Conference — ICLR 2025 Poster_

### Official Review · Reviewer_fu86 · 2024-11-01

**Soundness:** 2
**Presentation:** 2
**Contribution:** 2
**Rating:** 6
**Confidence:** 4

**Summary:**

This paper introduces ToolDial, a dataset specifically designed for Tool-Augmented Language Models (TALMs), focusing on simulating complex, real-world, multi-turn dialogues that involve external API usage. ToolDial consists of 11,111 dialogues, each structured to mirror realistic, multi-step user-system interactions where the model may need to request additional information or call relevant APIs in response to the user’s needs. The authors also propose a approach to dataset creation by using an API graph, which maps input-output compatibility between APIs. This structure enables TALMs to manage the sequence and dependencies of API calls within dialogues more effectively.

The paper evaluates various language models on ToolDial, providing valuable insights into their performance, particularly in handling multi-turn interactions requiring tool usage. This assessment highlights critical areas where TALMs could improve, notably in maintaining coherence and accuracy over extended dialogues.

**Strengths:**

(1) The ToolDial dataset addresses a gap in resources for TALMs, providing extensive and nuanced multi-turn dialogue interactions that are more reflective of real-world use cases.

(2) Dataset generation leverages an API graph with GPT models, reducing the reliance on human-curated data and offering a scalable approach that may inspire similar future methodologies.

(3) By evaluating multiple language models on ToolDial, the paper provides actionable insights into model performance, helping to identify specific challenges TALMs face in multi-turn, tool-augmented interactions.

**Weaknesses:**

1. Heavy reliance on GPT-generated dialogues introduces the risk of bias and synthetic errors inherent to the model, which could undermine the dataset’s realism.

2. The paper lacks a detailed error analysis, missing an opportunity to discuss specific areas where TALMs underperform and how ToolDial might be used to address these issues.

3. The evaluation process has potential reliability issues:
(1) There are concerns that the upper bound results (when using ground truth) are lower than expected, especially with the TD-Llama model, which performed worse than anticipated.
(2) TD-Llama (finetuned on Llama3-8b) scores lower than GPT-3.5-turbo in many aspect, a result that appears counterintuitive.

**Questions:**

1. The concept of "correctness" is unclear. Clearer definitions and criteria for correctness could improve result interpretability. It is calculated at the session level, right?

2. It is unclear how many test samples were used in Tables 4 and 5. The presence of integer scores is unusual, given that the test set consists of 1,166 samples. Including precise sample counts and calculation methods could enhance transparency.

3. It remains uncertain if DST or action prediction are essential to achieving high correctness scores. It seems models could achieve high correctness even with low DST or action prediction accuracy. The authors are encouraged to conduct an ablation study to determine the necessity and impact of DST and action prediction on overall performance.

4. The paper would benefit from a comparison with other works that utilize Tool graphs, such as “Taskbench: Benchmarking Large Language Models for Task Automation.”

---

> ### Author Response · Authors · 2024-11-20
>
> Thank you for your time and effort to review our paper. We sincerely appreciate the recognition of our efforts in addressing the resource gap for TALMs through the ToolDial dataset, which offers extensive multi-turn interactions reflective of real-world use cases. Additionally, thank you for recognizing the value of our dataset generation approach. By leveraging an API graph with GPT models, our scalable method reduces reliance on human-curated data and provides a framework that can inspire similar methodologies in the future.
>
> &nbsp;
>
> **Q1.** Heavy reliance on GPT-generated dialogues introduces the risk of bias and synthetic errors inherent to the model, which could undermine the dataset’s realism.
>
> **A1.** Thank you for highlighting this issue. When instructing GPT to generate multi-turn dialogues between a Tool-Augmented Language Model (TALM) and a user without any guidance, the resulting dialogues indeed tend to be overly repetitive and monotonous. Specifically, certain types of APIs are disproportionately preferred, and the actions performed by both the user and system lack variety, typically following a simple "inform intent - response" pattern. This results in an overly homogeneous set of situations.
>
> To avoid this issue, our data generation procedure incorporates several guiding strategies. First, we sampled various triples from the API graph to include a wider range of APIs and domains. Additionally, we developed dialogue scenarios based on 16 user and system actions and 23 action sequences derived from them, which were then integrated into the generated dialogues. Notably, when GPT generates dialogues, it often neglects to include turns where either the user or the system performs actions related to rejection or negation. To mitigate this, we introduced user actions like “fail_inform” and “negate”, and system actions like “response_fail”, allowing for a wider range of situations in the dialogue data.
>
> Moreover, although not mentioned in the paper, we observed that dialogues generated without guidance often exhibit consistent speaking styles for both the user and the system. For instance, the system’s “request” action often prompts "Can you give me 'param1' and 'param2'?". To introduce more variation, we defined several distinct speaking styles for these recurring actions, including:
> * **user action**
>    - inform
>       - Sure! ~
>       - Ok ~
>       - Certainly!
>     - affirm
>       - Yes, that works.,
>       - That would be great.
>       - Sure, that sounds good.,
>       - Yes, please proceed.
>     - negate
>       - No, that’s not what I meant.
>       - I'm good. Thank you though.
>       - Umm... that's not what I want..
> * **System action**
>     - request
>       - To call ~
>       - I need~,
>       - May I ask for ~,
>       - Please tell me ~,
>     - clarify
>       - Could you please provide more ~,
>       - I’m not sure I understand. Can you clarify ~,
>       - Could you explain that in more ~,
>       - Can you clarify your ~
> We incorporated a mechanism to randomly select from these predefined speaking styles during the scenario generation step (Section 3.3). For each action, we included a prompt to instruct GPT to generate utterances in line with the randomly selected style for the corresponding turn. We will add this description in our revision.

---

> ### Author Response · Authors · 2024-11-20
>
> **Q2.** The evaluation process has potential reliability issues: (1) There are concerns that the upper bound results (when using ground truth) are lower than expected, especially with the TD-Llama model, which performed worse than anticipated. (2) TD-Llama (finetuned on Llama3-8b) scores lower than GPT-3.5-turbo in many aspect, a result that appears counterintuitive.
>
> **A2.** Thank you for raising this issue. To answer your question, we examined our current method and identified a discrepancy in the prompt format used during training and inference for the Llama-series models compared to the GPT-based models. This discrepancy critically impacted the performance of the Llama models, leading to suboptimal results. To address this, we revised the prompt format for the Llama-based models to align with the format used for the GPT-based models. We also added more baselines as per the feedback of the reviewers. The updated scores are as follows:
>
> | Model Type              | Model Name                | DST (With GT) | DST (W/O GT) | Action (With GT) | Action (W/O GT) | Correctness (W/O GT) |
> |-------------------------|---------------------------|---------------|--------------|------------------|-----------------|--------------------------|
> | GPT Models (Zeroshot)   | GPT-3.5-turbo            | 38.8          | 33.1         | 53.5             | 54.1            | 95.4                     |
> |                         | GPT-4o-mini              | 58.8          | 67.7         | 63.7             | 60.2            | 96.6                     |
> |                         | GPT-4o                   | 81.4          | 67.8         | 57.6             | 63.7            | 96.7                     |
> |                         | GPT-4-turbo              | 77.5          | 68.6         | 64.2             | 61.5            | 97.1                     |
> | Open source LLMs  (Zeroshot)        | CodeLlama-7b-Instruct-hf | 47.2          | 28.9         | 35.7             | 30.0            | 81.7                     |
> |              | Qwen2.5-Coder-7B-Instruct| 48.9          | 34.2         | 55.8             | 46.8            | 93.9                     |
> |                         | Llama3-8b-Instruct       | 53.4          | 24.5         | 37.7             | 35.5            | 91.5                     |
> | Fine-tuning  | TD-llama                 | 92.7          | 72.2         | 77.5             | 91.0            | 88.4                     |
>
>
> The scores for the GPT-series models remain unchanged. However, the scores for vanilla Llama and TD-Llama have been changed.
> As shown in the DST scores in the table, we observed significant performance improvements for both Vanilla Llama and TD-Llama, surpassing the performance of GPT-3.5. We will update our revision with these updated scores in Table 4 and highlight the observed performance trends accordingly.

---

> ### Author Response · Authors · 2024-11-20
>
> **Q3.** The paper lacks a detailed error analysis, missing an opportunity to discuss specific areas where TALMs underperform and how ToolDial might be used to address these issues.
>
> **A3-1.** Thank you for your advice. The following are the results of our error analysis and we will include this in the appendix of our revision. The following analysis is based on the updated scores for TD-Llama and Llama Instruct as discussed in Q2 (not from the original Table 4).
>
> **DST Error analysis**
> |                | GPT-3.5-turbo |      | GPT-4o-mini |       | GPT-4-turbo |         | GPT-4o |         | Llama3-8b-inst |        | TD-Llama    |
> |---------------|-------|-----|--------|----------|--------|--------------|--------|--------------|--------------------|--------------|-------------|
> |                  | W GT | W/O GT | W GT | W/O GT  | W GT| W/O GT | W GT| W/O GT | W GT | W/O GT| W GT| W/O GT |
> | # of Error                | 4128    | 4512   | 2781  | 2177   | 1515  | 2117   | 1257 | 2169   | 3138| 5090   | 492  | 1873|
> | Generation Error    | 0 | 0 | 0 | 0 | 0 | 0| 0| 0| 3| 0| 260| 1619|
> | API Conf. Err (GT = T) | 1609 | 1841| 1060 | 504 | 211| 224 | 243| 1607| 583| 1014 | 30   | 1 |
> | API Conf. Err (GT = F)| 750  | 410 | 848| 373| 692| 891 | 343 | 133 | 723| 923| 0 | 0 |
> | Format Err| 532| 502| 153| 0| 0| 0| 74| 0| 531| 319| 61|103|
> | Slot Error| 1139 | 1674| 508| 912 | 430| 774 | 443| 221| 1101| 2663| 6| 23|
> | Value Error |561| 630| 398|823| 498| 626| 495| 221| 846| 1423| 134| 144|
>
> - The W/O GT results for TD-Llama could not be fully entered due to length constraints, so we will present here in text form. From Generation Error to Value Error, the results are 1873, 1619, 1, 0, 103, 23, and 144, respectively.
>
> - The DST error table shows error counts, where slot and value errors can overlap in a single prediction, causing their sum to exceed the total errors.
>
> For better understanding, we have provided a description of each error type on **Error Type**. By referring to the **Format of the dialogue state**, you will be able to better understand exactly which aspects each error type pertains to.
>
> **Format of the dialogue state** (More details are in Appendix.4 of the paper)
>
> - When there is no confirmed API:
>
>   - Dialogue State: {API confirmed: false, API status: none}
>
> - When the API is confirmed and some input parameter information can be extracted from dialogue history:
>
>   - Dialogue State: {API confirmed: true, API status: {API name: “API1”, Required parameters: {param1: “value1”, param2: “”}, Optional parameters: {param3: “value3”}}}
>
> Here, we call the **param1, param2, param3 as slots**, and **value1, value3 as values.**
>
> **Error Type**
>
> - **Generation Error**: This occurs when the dialogue state dictionary is not generated at all.
>
> - **API Confirmation Error (GT = True)**: This error happens when API is confirmed (api_confirmed=true), but is incorrectly predicted as it is confirmed (api_confirmed=false).
>
> - **API Confirmation Error (GT = False)**: This error occurs when the API is not confirmed (api_confirmed=false), but the model incorrectly predicts it as confirmed (api_confirmed=true).
>
> - **Format Error**: This occurs when the dialogue state does not fully generate all fields such as api_confirmed, api_status, required parameters, and optional parameters.
>
> - **Slot Error**: When api_confirmed is true, this error involves generating a dialogue state that does not include all required and optional parameter slots as specified in the API documentation.
>
> - **Value Error**: This error involves incorrectly extracting the slot’s value from the dialogue history, with the following types:
>
>   - **Extracting Input Value from Multiple Result Error**: This error occurs when an appropriate value cannot be selected from multiple results returned by the API output (as seen in turns 6 and 7 of Figure 2).
>
>   - **Inform Intent Add Error**: This occurs when there is a value within the user query that could be used as an input parameter (inform intent clear add), but the model fails to track it.
>
>    - **Other General Input Parameter Extraction Errors**: Errors that occur in typical situations where the input parameter is extracted incorrectly.
>
> **Result analysis for DST error**
> - Various errors appear in the GPT series and Llama3 Instruct models. In the W/O GT setting, which resembles real-world scenarios, more errors occur.
> - In TD-Llama, errors decrease significantly, though generation errors remain common. Other frequent errors include Extracting Input Value from Multiple Results and Inform Intent Add errors.
> - To reduce errors through fine-tuning, a simple approach to address "generation errors" can be proposed. However, "Extracting Input Value from Multiple Result Error," involving noisy lists, remains a challenging problem and an important future direction.

---

> ### Author Response · Authors · 2024-11-20
>
> **Q3.** The paper lacks a detailed error analysis, missing an opportunity to discuss specific areas where TALMs underperform and how ToolDial might be used to address these issues.
>
> **A3-2.** For the action prediction, we present an F1-score table as part of the error analysis.
>
> **F1-score table**
> | Setting | Model               | response | responsefail | request | retrievecall | clarify | systembye | suggest | call |
> |---------|---------------------|----------|--------------|---------|--------------|---------|-----------|---------|------|
> | **With GT**    | GPT-3.5-turbo              | 63.8     | 0.0          | 28.4    | 66.2         | 1.3     | 95.5      | 0.0     | 53.4 |
> |         | GPT-4o-mini          | 78.9     | 0.0          | 44.3    | 67.4         | 64.5    | 97.2      | 0.0     | 67.0 |
> |         | GPT-4-turbo        | 93.6     | 0.0          | 18.1    | 87.5         | 56.7    | 97.2      | 29.9    | 56.4 |
> |         | GPT-4o             | 88.3     | 0.0          | 13.7    | 74.9         | 29.6    | 97.2      | 24.6    | 54.1 |
> |         | Llama3-8b-Instruct | 46.4     | 0.0          | 8.5     | 23.7         | 0.0     | 99.8      | 14.0    | 44.4 |
> |         | TD-Llama           | 100.0    | 77.5         | 44.8    | 97.2         | 77.4    | 99.9      | 16.8    | 68.6 |
> | **W/O GT**   | GPT-3.5-turbo              | 70.7     | 0.0          | 1.3     | 77.6         | 0.0     | 93.0      | 0.0     | 49.7 |
> |         | GPT-4o-mini          | 88.5     | 0.0          | 36.1    | 62.6         | 0.0     | 97.2      | 0.0     | 65.1 |
> |         | GPT-4-turbo        | 96.6     | 0.0          | 10.8    | 79.9         | 40.6    | 97.2      | 35.5    | 57.8 |
> |         | GPT-4o             | 95.8     | 0.0          | 14.3    | 81.2         | 38.6    | 97.2      | 46.1    | 62.0 |
> |         | Llama3-8b-Instruct | 30.5     | 0.0          | 1.9     | 27.3         | 0.0     | 93.1      | 9.4     | 42.0 |
> |         | TD-Llama           | 98.2     | 99.1         | 78.4    | 94.5         | 99.8    | 100.0     | 99.9    | 86.9 |
>
> The following analysis is based on the updated scores for TD-Llama and Llama Instruct as discussed in Q2 (not from the original Table 4). We will update Table 5 according to the **F1-score table** in our revision.

---

> ### Author Response · Authors · 2024-11-20
>
> **Q4.** The concept of "correctness" is unclear. Clearer definitions and criteria for correctness could improve result interpretability. It is calculated at the session level, right?
>
> **A4.** We apologize for the unclarity of the term “correctness.” One of the key capabilities of TALM is to provide responses to the user that are faithful to the API results without any hallucination. The metric Correctness measures this by comparing the utterance following the system's response action (the point at which it replies to the user based on API call results). Scoring is conducted using G-Eval with GPT model, where a score of 1 is assigned if the system's response aligns with the API call results, and 0 if it does not.
>
> We are considering renaming this metric from _Correctness_ to _Hallucination Ratio_ or _Faithfulness_ to make the metric name more intuitive.

---

> ### Author Response · Authors · 2024-11-20
>
> **Q5.** It is unclear how many test samples were used in Tables 4 and 5. The presence of integer scores is unusual, given that the test set consists of 1,166 samples. Including precise sample counts and calculation methods could enhance transparency.
>
>
> **A5.** The current test set consists of 1,166 dialogues. Among these, there are 6,746 dialogue states recorded in the middle of dialogues, and 9,200 actions. Therefore, **6,746 instances were used for DST evaluation**, and **9,200 for action prediction.** For correctness (i.e., the hallucination metric), which assesses the system’s response utterance after the “response” action, **943 dialogues  were evaluated for correctness**, excluding those that end with a “response fail” from the total 1,166 dialogues. Additionally, we will present all experimental scores to the third decimal place.
>
> The metrics are calculated as follows:
> - **DST (Dialogue State Tracking)**: Each turn’s label dialogue state dictionary and predicted dialogue state dictionary were converted to lowercase, and special characters were removed. The evaluation was based on whether the two dialogue states matched completely.
> - **Action Prediction**: Each turn’s label action and predicted action were converted to lowercase, and special characters were removed. Evaluation was based on whether they matched exactly.
> - **Correctness**: Following the same method as G-Eval, a GPT model with temperature set above 0 evaluated each response (a total of 943) 10 times. The average of the 10 results (all either 0 or 1) was used as the score.
> We will add this clarification in our revision.

---

> ### Author Response · Authors · 2024-11-20
>
> **Q6.** It remains uncertain if DST or action prediction are essential to achieving high correctness scores. It seems models could achieve high correctness even with low DST or action prediction accuracy. The authors are encouraged to conduct an ablation study to determine the necessity and impact of DST and action prediction on overall performance.
>
> **A6.** We believe that this confusion comes from the original unclarity of the term “correctness”. We have clarified the term in Q4. In short, “Correctness” assesses whether the system generates a response that aligns with the output of an API call. Therefore, the correctness score is unrelated to DST or action prediction scores. Please let us know if you have further questions about this.
> Additionally, you have asked about _overall performance_. If you could clarify what specifically you mean by this, we would be glad to provide an answer to that.
>
> For now, we interpreted “overall performance” as a model’s ability to generate a suitable thought, action, dialogue state, and message as a reply, given the dialogue history and the most recent user utterance, all in a single generation step. To assess this capability, we designed an experiment to evaluate TD-llama's performance. In this experiment, reasoning traces were generated for a total of 5,213 user utterances from test dialogues. If both the action and dialogue state within each reasoning step were accurately generated, the result was marked as True; otherwise, it was marked as False. This evaluation yielded a performance score of **76.6%**. Additionally, for 1,166 test dialogues, we measured the rate at which the reasoning trace for all turns was correctly generated from the first to the last, resulting in an accuracy rate of approximately **57.1%**.

---

> ### Author Response · Authors · 2024-11-20
>
> **Q7.** The paper would benefit from a comparison with other works that utilize Tool graphs, such as “Taskbench: Benchmarking Large Language Models for Task Automation.”
>
> **A7.** Thank you for introducing this relevant paper. To briefly summarize the similarities and differences between ToolDial and TaskBench:
>
> **Similarities**
> - Both papers address generating dialogue datasets that reflect tool dependency scenarios.
> - Both constructed a tool graph to represent tool dependency information, and used extracted subgraphs to generate dialogues.
>
>
> **Differences**
> - TaskBench generates scenarios where a single turn can be solved by chaining tools based on the extracted subgraph. In contrast, ToolDial generates scenarios where the agent determines parameter values through interaction with the user over multiple turns. The system either requests input parameter information from the user or, if the user cannot provide it, finds and calls other APIs to obtain the necessary information. TaskBench scenarios, however, immediately chain tools without involving the user. We believe that this difference is crucial because API calls are often costly and slow, and it is more natural to ask users for information they already have.
>
> - The evaluation of TaskBench focuses solely on assessing the system’s ability for proper tool selection and tool chain usage. In contrast, ToolDial is “dialogue data” based on real-world conversational scenarios, encompassing 16 user and system actions. Its evaluation emphasizes the skills needed to address user queries within multi-turn dialogue contexts. In both works, an API graph is essential, but obtaining this graph itself is a challenging problem. In our paper, we propose and evaluate a new method to construct an API graph, whereas TaskBench does not specify whether its graph construction is manual or automated.
>
>
> We will discuss these points in the Related Works section of our paper.

---

> ### Author Response · Authors · 2024-11-22
>
> Thank you again for your time and effort to review our paper and provide constructive feedback. We have made our best effort to incorporate your suggestions and address your questions in our rebuttal. We understand you're busy but we would greatly appreciate it if you could review our rebuttal and let us know if it addresses your feedback and concerns.

---

> ### Author Response · Authors · 2024-11-25
>
> Dear Reviewer fu86, we apologize for urging you again to engage with our rebuttal. As the discussion phase ends in less than two days, we hope to ensure sufficient time to discuss with you to address your concerns thoroughly. We have made our best effort to respond to your questions and have incorporated all your suggestions and feedback into our revised submission (now uploaded). Below is a summary of our rebuttal, with references to the specific sections in the revision where each point has been addressed:
>
> 1. **Dataset realism and GPT biases:** We acknowledged the risks of GPT-generated dialogues and detailed the strategies implemented in ToolDial to diversify scenarios and reduce repetitive patterns. These include sampling diverse API triples and defining specific user/system actions and speaking styles, enhancing the dataset’s realism and variability. (Section 3.4, “Model Biases”)
>
> 2. **Evaluation and model performance discrepancies:** We identified a prompt format issue affecting Llama-based models and presented updated performance scores. The revised results demonstrate significant improvements for TD-Llama, now surpassing GPT-3.5. Additional baseline models were also included based on other reviewers’ suggestions. (Section 4.3, Tables 4 & 5)
>
> 3. **Error analysis:** We provided a detailed breakdown of error types and counts, highlighting challenges like extracting values from noisy API outputs. F1 scores for action prediction were also included for better contextualization. (Appendix A.9 and Section 4.3)
>
> 4. **Clarification of correctness metric:** We clarified that "Correctness" measures response faithfulness to API results and is unrelated to DST or action prediction scores. To improve clarity, we renamed this metric “faithfulness.” (Section 4.1)
>
> 5. **Sample counts and metric definitions:** We clarified details of the test samples (e.g., 1,166 dialogues, 6,746 DST instances, and 9,200 action predictions) and provided precise calculation methods. Scores have been updated to three decimal places. (Section 4.1)
>
> 6. **Overall performance:** We addressed concerns about the relationship between DST/action prediction and correctness, and shared an additional experiment evaluating TD-Llama’s “overall performance,” achieving 76.6% accuracy for individual turns and 57.1% for entire dialogues. (Section 4.3, “Overall Performance”)
>
> 7. **Comparison with TaskBench:** We outlined the similarities and differences between TaskBench and ToolDial, emphasizing ToolDial's focus on multi-turn dialogues and user-agent interactions, unlike TaskBench's tool-chaining approach. (Section 2)
>
> Beyond the questions and suggestions outlined above, we have also addressed feedback from other reviewers. This includes adding more baseline models, providing statistics on the domains covered in ToolDial, adding a running example to illustrate dataset construction, clarifying the novelty of our work and its contribution to ICLR, and making structural adjustments between the main text and appendix. These updates have been well-received by the reviewers.
>
> We appreciate your engagement and remain ready to address any further concerns. Thank you!

---

> ### Comment · Reviewer_fu86 · 2024-11-26
> **comments**
>
> The author has addressed most of my concerns in the revised version, which allows me to raise my score accordingly.
>
> However, I must highlight a significant difference between the results reported in the new version and those in the original submission. Specifically, the DST results for TD-llama with ground truth (GT) have shown an unprecedented increase from **25 to 92.7**. This dramatic improvement raises questions about consistency and reproducibility that warrant careful consideration.
>
> Please take this substantial gap into account when making the final decision.

---

> > ### Author Response · Authors · 2024-11-27
> > **Dear Reviewer fu86**
> >
> > Thank you for your thoughtful feedback and for updating your score based on our rebuttal. We acknowledge your concerns about consistency and reproducibility and would like to clarify that such issues are not present for the following reason.
> >
> > The primary reason for this improvement lies in a straightforward modification to the prompt format. Specifically, in the original submission, the prompt format for TD-llama included approximately 16 special tokens, which inadvertently imposed a significant burden on the model. This design required the model to allocate resources to learning the representation of these new tokens rather than focusing on its reasoning capabilities. As a result, TD-llama in the DST-with-GT setting often failed to generate dialogue states in the correct format.
> >
> > In the revised submission, informed by valuable reviewer feedback, we adopted the same prompt format used with GPT-based models, removing the special tokens. This adjustment allowed the model to fully leverage its reasoning capabilities, leading to the significant improvement observed. To ensure reproducibility, we have detailed the revised prompt format in Table 18 and can provide the original prompt format upon request.
> >
> > Additionally, we conducted a systematic error analysis of the new results to further validate our findings and have presented these insights transparently in Section A.9 in our revision.
> >
> > We hope this explanation alleviates any concerns regarding consistency or reproducibility. Thank you again for your constructive feedback, which has been instrumental in strengthening our work.

---

### Official Review · Reviewer_qYBo · 2024-11-04

**Soundness:** 3
**Presentation:** 3
**Contribution:** 3
**Rating:** 6
**Confidence:** 5

**Summary:**

The paper addresses the limitations of existing benchmarks for Tool-Augmented Language Models (TALMs), which often feature simplistic, single-turn dialogues that don't reflect real-world complexities. To bridge this gap, the authors introduce ToolDial, a dataset comprising 11,111 multi-turn dialogues with an average of 8.95 turns per dialogue, based on APIs from RapidAPI. ToolDial is designed to simulate rich user-system interactions by incorporating 16 types of user and system actions. The dataset also includes scenarios where the TALM must proactively seek additional APIs when the user fails to provide necessary information. To create ToolDial, the authors develop a method for generating an API graph that represents input and output compatibility between APIs, enabling sequential API calls. They then evaluate various language models on tasks like predicting correct actions, selecting appropriate APIs, and extracting input parameters from dialogue history. The results indicate that modern language models still have significant room for improvements.

**Strengths:**

1. The authors make appropriate citations and discussions to position their work and justify their contributions.

2. This paper makes solid contributions to the dialogue community. Two of the most popular dialogue state tracking (user intent extractions) are MWOZ and SGD. Both of them are created over 4 years ago. This is a timely work to establish a new benchmark. It also highlights many-turn interactions which differentiates itself from other API benchmark (e.g. API bank)

**Weaknesses:**

1. Although I recognize the contributions this paper makes to advance the development of dialogue systems, I am not sure the proposed methodologies for creating the dataset are adequately novel or aligned with the values of ICLR. Maybe the authors can provide more evidence and arguments in the rebuttal.

2. Lack of details about the dialogues and APIs: I would like to (1) see more descriptions of how complex the conversations are, as this is something the authors claim as a novelty, and (2) have a clear view of the APIs—what domains are you using and what types of arguments are you employing? Is there any fundamental difference between yours and the SGD?

**Questions:**

In Figure 2, the agent response contains symbolic terms from the API (seasonId and tournamentId). I am curious about the reasons for this design. Does this mean you are also showing the APIs to the users at the same time? This will make the interactions less natural.

I also have a writing suggestion for the authors. Try to condense the paper and move some essential information from Appendix to the main paper. I am constantly scrolling to Appendix section when reading the paper.

---

> ### Author Response · Authors · 2024-11-19
>
> Thank you for your time and effort to review our paper. We deeply appreciate the positive recognition of our work, particularly for appropriately positioning our contributions through citations and discussions, and for establishing a timely benchmark in the dialogue community. Also, thank you for recognizing our efforts to create a benchmark that stands out from existing ones such as MWOZ, SGD, and API Bank.
>
> &nbsp;
>
>
> **Q1.** Although I recognize the contributions this paper makes to advance the development of dialogue systems, I am not sure the proposed methodologies for creating the dataset are adequately novel or aligned with the values of ICLR. Maybe the authors can provide more evidence and arguments in the rebuttal.
>
> **A1.** Thank you for your question.
> Autonomous agents and their tool use are a significant research focus at ICLR, with several related studies, such as ToomLLM and ToolBench, being published. Advancing this field requires a benchmark dataset that reflects realistic and complex scenarios of user-agent interactions. Since most existing benchmarks are relatively simplistic; our work introduces ToolDial, which accounts for multi-turn dialogues, agent proactivity (see the next paragraph), diverse user and system actions, and real-world APIs with noisy API documentation from RapidAPI.
>
> A key focus of our work is on dialogues where the agent proactively calls multiple APIs in sequence. In real-world scenarios, it is common for users to fail to provide a parameter value requested by the agent. In such cases, the agent is expected to proactively search for and call another API to retrieve the necessary information, rather than simply stating that the user request cannot be completed. Generating dialogues that account for such agent proactivity is challenging because it requires identifying which APIs can provide information that can serve as input for other APIs. To address this, we developed an automated method for API graph construction and assessed its reliability. This graph facilitates the generation of dialogues with API dependencies and could potentially be utilized for API retrieval in future research.
>
> We believe that tool-augmented language models (TALMs) and traditional dialogue management (DM) systems share many similarities, but recent TALM research has somewhat overlooked the rich literature on DM systems. At the core of the DM research are dialogue state tracking and diverse user and system actions, which serve as explicit and interpretable methods for tracking an agent’s progress in task completion. Our work aims to bridge this gap by introducing these concepts into TALMs, thereby enriching the future research in the TALM field.

---

> ### Author Response · Authors · 2024-11-19
>
> **Q2.** Lack of details about the dialogues and APIs: I would like to (1) see more descriptions of how complex the conversations are, as this is something the authors claim as a novelty, and (2) have a clear view of the APIs—what domains are you using and what types of arguments are you employing? Is there any fundamental difference between yours and the SGD?
>
> **A2.**
>
> **(1)** Our dataset includes a total of 16 actions, combining both user and system actions, as shown in Table 1. This significantly surpasses the diversity of actions in other benchmark datasets, allowing us to capture a wider variety of real-world situations. Existing TALM benchmarks are typically limited to a few actions, such as the user’s “inform intent” and “inform action”, and the system’s “request” and “response” actions. However, in our dataset, we also consider the user’s action of “inform intent vague” and other actions like the user’s “fail inform”, “affirm”, and “negate” actions, as well as system actions like “clarify”, “suggest”, and “response fail”. Thanks to many combinations of these actions, ToolDial encompasses scenarios not covered in previous benchmarks and demonstrates novelty.
>
> Furthermore, the same type of action, such as the user’s “fail inform” or the system’s “request” can lead to different conversational flows depending on the specific situation in that turn. For instance, suppose the user fails to inform a requested parameter value. The subsequent dialogue may vary depending on whether the user simply states that they do not know the requested value or instead still provides other parameter values they can (as shown in the 3th turn of Figure 2). Our dataset accounts for both scenarios.
>
> Additionally, there are two types of request actions. The first is the typical action of asking the user for an input parameter. The second occurs when an API output returns multiple values in a list, prompting the system to ask the user to select one of these values (as shown in the 6th turn of Figure 2).
>
> In summary, unlike most existing benchmarks that are typically structured with only inform intent → request → inform → response actions, our dataset introduces a broader range of actions. Moreover, by varying the way the same actions are performed, our dataset provides much more complex scenarios than those in conventional datasets, highlighting the novelty of our work.
>
> **(2-1)** We used APIs from 28 domains: \
> _['Business', 'Data', 'Energy', 'Finance', 'Food', 'Business_Software', 'Database', 'Entertainment', 'Events', 'Health_and_Fitness', 'Location', 'SMS', 'Science', 'Social', 'Mapping', 'Music', 'Search', 'Logistics', 'Media', 'News_Media', 'Sports', 'Travel', 'Weather', 'Transportation', 'Financial', 'Gaming', 'Payments', 'Commerce']_
>
> Data (21.6%), Sports (18.7%), Weather (16.0%), Media (10.1%), Location (7.0%) make up the majority, with the remaining 23 domains evenly distributed.
> The input arguments of the APIs include approximately 399 parameters in the dataset, covering both required and optional parameters. These input parameters vary widely, with the most frequently appearing types being IDs, codes, coordinates, location names, languages, and dates. The data types of these arguments include string, bool, list, date, json, number, and others.
>
> **(2-2)** One of the main differences between ToolDial and SGD is that ToolDial addresses situations involving API dependencies, or "API call chains." Specifically, the user may be unable to provide an argument required to execute an API, and the agent proactively searches for and calls another API to retrieve the necessary information. In contrast, SGD does not cover such agent proactivity, and its API chains are relatively straightforward (e.g., find a restaurant -> book a restaurant). Furthermore, ToolDial encompasses more than 400 real-world APIs from RapidAPI, compared to just 44 APIs (or intents) in similar domains (e.g., Flights and Train, Hotels and Restaurants) covered by SGD. Generating complex dialogues based on a noisy pool of APIs, as in ToolDial, is inherently more challenging than in SGD, and thus we developed an automated method for API graph construction.
>
> In addition, while the primary motivation of SGD is on evaluating a model's domain transfer performance in unseen domains that are similar but slightly different (e.g., the model is trained on the Train domain and tested on the Flight domain), ToolDial is grounded in a broader range of APIs with noisy API documentation used in real-world scenarios. As a result, ToolDial offers a more realistic setting for both training tool-augmented models and evaluating their generalizability.

---

> ### Author Response · Authors · 2024-11-19
>
> **Q3.** In Figure 2, the agent response contains symbolic terms from the API (seasonId and tournamentId). I am curious about the reasons for this design. Does this mean you are also showing the APIs to the users at the same time? This will make the interactions less natural.
>
>
> **A3.** Thank you for highlighting this issue. When the system requests a parameter value from the user, we may either use the original parameter name as-is (e.g., tournamentId) or a more intuitive description, depending on the complexity of the parameter name.
>
>
> For instance, some API parameters in RapidAPI are not immediately clear based on their names alone (e.g., the parameter “X-User-Agent” expects a device type, such as “mobile” or “desktop”), which could confuse the user if requested by their names. To address this, during the Dialogue Scenario Generation step (Section 3.3), we included both the parameter name and its description in the “request” action instruction. This enables GPT to generate request utterances that are more user-friendly. For example, when the “X-User-Agent” parameter is required, our dataset prompts a response like “Please provide the device type you are using,” instead of “Please provide the X-User-Agent”.
>
> In Figure 2, specifically, symbolic terms (seasonId and tournamentId) are used because GPT judged their names to be intuitive enough for users to understand in context. However, when there is a substantial difference between a parameter’s name and its description, as in the X-User-Agent example, the request utterance is generated with greater reliance on the parameter's description.

---

> > ### Comment · Reviewer_qYBo · 2024-11-24
> > **Response to Author Rebuttal**
> >
> > I appreciate the authors make concrete efforts to address my questions and comments.
> > The explanations are examples are clear and support their claims in the paper. I am adjusting my scores. No further questions!

---

> > > ### Author Response · Authors · 2024-11-27
> > > **Dear Reviewer qYBo**
> > >
> > > Thank you for your feedback and for updating your score to reflect our efforts. We sincerely appreciate you highlighting the complexity of the dialogue scenarios, the diversity of the APIs, as well as the symbolic terms. We will work on updating the dataset creation section to ensure these aspects are more clearly presented.
> > >
> > > Once again, thank you for your constructive feedback.

---

> ### Author Response · Authors · 2024-11-19
>
> **Q4.** I also have a writing suggestion for the authors. Try to condense the paper and move some essential information from Appendix to the main paper. I am constantly scrolling to Appendix section when reading the paper.
>
> **A4.** Thank you for your suggestion. We plan to strengthen the main text by moving part of the explanation of user and system actions from Appendix A.3 to Section 3.2. Additionally, we plan to incorporate part of the explanation of dialogue state and retriever status from Appendix A.4 to Section 3.3. We would greatly appreciate it if you could provide more concrete suggestions that can strengthen our paper.

---

> ### Author Response · Authors · 2024-11-22
>
> Thank you again for your time and effort to review our paper and provide constructive feedback. We have made our best effort to incorporate your suggestions and address your questions in our rebuttal. We understand you're busy but we would greatly appreciate it if you could review our rebuttal and let us know if it addresses your feedback and concerns.

---

> > ### Comment · Reviewer_4sGc · 2024-11-23
> > **Response to the rebuttal**
> >
> > Thank you for the detailed response and the new experiments
> > I am updating my score to validate the efforts for the new experiments
> >
> > I appreciate the thoughtfulness of including several checks to make the dataset as natural as possible. I will leave it up to the broader community to judge the merits of bench marking LLMs with LLM generated datasets.
> > Please update the dataset creation section with the new points discussed in the rebuttal in the next draft of this work.
> >
> > Re benchmarking on non LLM based baselines, as this work is primarily a dataset introduction, it would be useful to judge if the proposed dataset is complex for LLMs alone or complex as a problem in general. I will leave it to your better judgement and time if you want to add these new experiments.
> >
> > Lastly, in future rebuttals, please use fewer comment boxes, perhaps condense different questions per comment box. Getting 34 email notifications is not ideal.

---

> > > ### Author Response · Authors · 2024-11-24
> > > **Dear Reviewer 4sGc**
> > >
> > > Thank you for your feedback and for updating your score to reflect our efforts. We appreciate your acknowledgment of the dataset’s naturalness and your insights on benchmarking and complexity. We will update the dataset creation section with the new points in the next draft and carefully consider including additional experiments. Your feedback has been invaluable in improving our work.
> > >
> > > We also apologize for the inconvenience caused by the excessive number of email notifications and will ensure that responses are consolidated in future rebuttals to avoid similar issues.

---

### Official Review · Reviewer_4sGc · 2024-11-04

**Soundness:** 3
**Presentation:** 3
**Contribution:** 3
**Rating:** 8
**Confidence:** 3

**Summary:**

This work introduces a new dataset for tool calling in LLMs for a multi-turn setup. It consists of conversations where the API calls are dependent on the context in the previous turns as well as require the system to request/clarify more information from the user. The dataset has been generated. The benchmark is evaluated on three tasks - action prediction, dialogue state tracking, and correctness of the API call. Experiments are conducted on four GPT-based versions and one fine-tuned LLaMA.

Score was updated from 6-> 8. See comments for the updated score

**Strengths:**

1. Moving to multi-turn use of tool calling is new and timely to the TALM literature.
2. Instead of relying on one task for evaluation, the benchmark supports three smaller tasks to get a better understanding of the model's performance.

**Weaknesses:**

1. Non-LLM based baselines need to be included for completeness

2. The dataset was constructed using an LLM (GPT4-o-mini) - perhaps some artifacts in generation can lead to undesirable effects . The authors back up the dataset quality through human evaluation as well, but perhaps a blind test prepared by human authors can also be helpful.

3. The experiments in this work are still preliminary (see suggestions). Addition of more experimental details, clearly signifying the type of prompting baselines, inclusion of another open source LLM, experimenting with chain-of-thought like prompting (https://arxiv.org/abs/2403.04656) can also be useful.

**Questions:**

Questions:
1. What was the setting for the GPT-based experiments - zero-shot or in-context learning? Please add these details in the next version of the draft
2. Could you explain why does GPT-4o-mini perform better without ground truth in Table 4?
What are the domains of the examples in the dataset?


Suggestions:

1. Please mention the language of the dataset in the paper
2. Please include qualitative examples from the dataset describing (i) examples after the data collection is complete and (ii) failures of respective LLMs
3. Please report at least three significant digits when reporting the results
4. As it is a multi-turn dataset, inclusion of task completion metric - when the model performs all the correct api calls at every turn it is marked as a complete task. See goal accuracy under https://aclanthology.org/2022.acl-short.35
5. Please include vanilla LLaMa results in Table 5.
6. While GPT based results are not reproducible (based on API changes), it may be useful to release the predictions from those runs for future work to be built on this work.
7. While it is exciting to benchmark the models on, non LLM based baselines such as BERT based/T5 based dialogue state tracking/action recognition should also be evaluated for completeness
8. The dataset construction section is a little hard to understand. Perhaps a running example may make the section clear.
9. [Future work] As API calls are similar to function calling in Python, it would be useful to also benchmark CodeLLaMa/finet-uned CodeLLaMa for the given benchmark

Missing References:

Léo Jacqmin, Lina M. Rojas Barahona, and Benoit Favre. 2022. “Do you follow me?”: A Survey of Recent Approaches in Dialogue State Tracking. In Proceedings of the 23rd Annual Meeting of the Special Interest Group on Discourse and Dialogue, pages 336–350, Edinburgh, UK. Association for Computational Linguistics.

Schick, Timo, Jane Dwivedi-Yu, Roberto Dessì, Roberta Raileanu, Maria Lomeli, Luke Zettlemoyer, Nicola Cancedda and Thomas Scialom. “Toolformer: Language Models Can Teach Themselves to Use Tools.” ArXiv abs/2302.04761 (2023): n. pag.


Nikita Moghe, Patrick Xia, Jacob Andreas, Jason Eisner, Benjamin Van Durme, and Harsh Jhamtani. 2024. Interpreting User Requests in the Context of Natural Language Standing Instructions. In Findings of the Association for Computational Linguistics: NAACL 2024, pages 4043–4060, Mexico City, Mexico. Association for Computational Linguistics.

Please see the related work of:


Qin, Yujia, Shi Liang, Yining Ye, Kunlun Zhu, Lan Yan, Ya-Ting Lu, Yankai Lin, Xin Cong, Xiangru Tang, Bill Qian, Sihan Zhao, Runchu Tian, Ruobing Xie, Jie Zhou, Marc H. Gerstein, Dahai Li, Zhiyuan Liu and Maosong Sun. “ToolLLM: Facilitating Large Language Models to Master 16000+ Real-world APIs.” ArXiv abs/2307.16789 (2023): n. pag.

---

> ### Author Response · Authors · 2024-11-20
>
> Thank you for your time and effort to review our paper. We sincerely appreciate the positive feedback on our paper, highlighting the novelty and timeliness of introducing multi-turn tool calling to the TALM literature and the value of our benchmark in offering a more comprehensive evaluation through three smaller tasks instead of relying on a single one.
>
> &nbsp;
>
>
> **Q1.** Non-LLM based baselines need to be included for completeness
>
> **A1.** Thank you for your suggestion. Although we haven't included non-LLM baselines yet, we have added experiment results for more baselines, including **CodeLlama-7b-Instruct-hf, Qwen2.5-Coder-7B-Instruct** as per the feedback of the reviewers as shown below.
>
> We first want to clarify that upon reviewing our original results based on Reviewer 3’s feedback, we found that the prompt format used during the inference of vanilla Llama and the fine-tuning and inference of TD-Llama was highly suboptimal. Therefore, we re-ran the experiments for the Llama3 series models using the same prompt format as in the ChatGPT model experiments. The final results are as follows.
>
> | Model Type              | Model Name                | DST (With GT) | DST (W/O GT) | Action (With GT) | Action (W/O GT) | Correctness (W/O GT) |
> |-------------------------|---------------------------|---------------|--------------|------------------|-----------------|--------------------------|
> | GPT Models (Zeroshot)   | GPT-3.5-turbo            | 38.8          | 33.1         | 53.5             | 54.1            | 95.4                     |
> |                         | GPT-4o-mini              | 58.8          | 67.7         | 63.7             | 60.2            | 96.6                     |
> |                         | GPT-4o                   | 81.4          | 67.8         | 57.6             | 63.7            | 96.7                     |
> |                         | GPT-4-turbo              | 77.5          | 68.6         | 64.2             | 61.5            | 97.1                     |
> | Open source LLMs  (Zeroshot)        | CodeLlama-7b-Instruct-hf | 47.2          | 28.9         | 35.7             | 30.0            | 81.7                     |
> |              | Qwen2.5-Coder-7B-Instruct| 48.9          | 34.2         | 55.8             | 46.8            | 93.9                     |
> |                         | Llama3-8b-Instruct       | 53.4          | 24.5         | 37.7             | 35.5            | 91.5                     |
> | Fine-tuning  | TD-llama                 | 92.7          | 72.2         | 77.5             | 91.0            | 88.4                     |
>
>
> The scores for the ChatGPT series models remain the same. However, the scores for vanilla Llama and TD-Llama have changed.
> We will update Table 4 accordingly in our revision.

---

> ### Author Response · Authors · 2024-11-20
>
> **Q2.** The dataset was constructed using an LLM (GPT4-o-mini) - perhaps some artifacts in generation can lead to undesirable effects . The authors back up the dataset quality through human evaluation as well, but perhaps a blind test prepared by human authors can also be helpful.
>
> **A2.** Thank you for your suggestion. When instructing GPT to generate multi-turn dialogues between a Tool-Augmented Language Model (TALM) and a user without any guidance, the resulting dialogues indeed tend to be overly repetitive and monotonous. Specifically, certain types of APIs are disproportionately preferred, and the actions performed by both the user and system lack variety, typically following a simple "inform intent - response" pattern. This results in an overly homogeneous set of situations.
>
> To avoid this issue, our data generation procedure incorporates several guiding strategies. First, we sampled various triples from the API graph to include a wider range of APIs and domains. Additionally, we developed dialogue scenarios based on 16 user and system actions and 23 action sequences derived from them, which were then integrated into the generated dialogues. Notably, when GPT generates dialogues, it often neglects to include turns where either the user or the system performs actions related to rejection or negation. To mitigate this, we introduced user actions like “fail_inform” and “negate”, and system actions like “response_fail”, allowing for a wider range of situations in the dialogue data.
>
> Moreover, although not mentioned in the paper, we observed that dialogues generated without guidance often exhibit consistent speaking styles for both the user and the system. For instance, the system’s “request” action often prompts "Can you give me 'param1' and 'param2'?". To introduce more variation, we defined several distinct speaking styles for these recurring actions, including:
>
> * **user action**
>    - inform
>       - Sure! ~
>       - Ok ~
>       - Certainly!
>     - affirm
>       - Yes, that works.,
>       - That would be great.
>       - Sure, that sounds good.,
>       - Yes, please proceed.
>     - negate
>       - No, that’s not what I meant.
>       - I'm good. Thank you though.
>       - Umm... that's not what I want..
> * **System action**
>     - request
>       - To call ~
>       - I need~,
>       - May I ask for ~,
>       - Please tell me ~,
>     - clarify
>       - Could you please provide more ~,
>       - I’m not sure I understand. Can you clarify ~,
>       - Could you explain that in more ~,
>       - Can you clarify your ~
>
> We incorporated a mechanism to randomly select from these predefined speaking styles during the scenario generation step (Section 3.3). For each action, we included a prompt to instruct GPT to generate utterances in line with the randomly selected style for the corresponding turn.
>
> The blind test you suggested is an interesting experiment, and we leave it to future work. For now, we believe that generated dialogues do not necessarily need to be indistinguishable from human-authored dialogues. Human-authored task-oriented dialogues are often error-prone (e.g., MultiWOZ has undergone multiple revisions due to errors) and tend to lack diversity (e.g., SGD proposed an automated dialogue generation method to address this limitation). We made significant efforts to mitigate potential issues arising from GPT's biases, as evidenced by the satisfactory scores achieved in both G-Eval and human evaluations.

---

> ### Author Response · Authors · 2024-11-20
>
> **Q3.** The experiments in this work are still preliminary (see suggestions). Addition of more experimental details, clearly signifying the type of prompting baselines, inclusion of another open source LLM, experimenting with chain-of-thought like prompting can also be useful.
>
>
> **A3.** Thank you for your suggestions. The baseline prompt used in our experiments was based on ReAct, with each reasoning step comprising a thought, an action, and various types of observations (e.g., retriever status, API documentation, call result). The reasoning step includes the process of thought-action based on ReAct before the system replies to the user’s utterance. Below is the prompt baseline used in the experiments.
>
> ```
> User: Can you provide insights into which articles on NYTimes.com have been shared the most on social media platforms, particularly Facebook, to understand active engagement trends?
> System:
> - Thought: The user seems to have intent. I will call the retriever.
> - Action:retriever_call
> - Retriever status: {'retriever_call': 'true', 'retrieved_api': {'Business|ny_times_most_popular|/shared/{period}/{share_type}.json': 0.71372604, ….}
> - API documentations: {'api_name': 'Business|ny_times_most_popular|/shared/{period}/{share_type}.json', .....}
> - Thought: Based on results of the retriever, I need to call Business|ny_times_most_popular|/shared/{period}/{share_type}.json. To call it, I need to ask 'share_type' and 'period' to the user.
> - Action:request
> - Retriever status: {'retriever_call': 'false', 'retrieved_api': 'none'}
>  - Message: Could you please provide the type of sharing on Facebook for articles and the time period for retrieving popular articles?
> - Dialogue State: <Current dialogue state>
> User: Certainly! The share type is Facebook and the period is 7 days.
> System:
> - Dialogue State: <Current dialogue state>
> - Thought: Based on the user's response, now I can call the API Business|ny_times_most_popular|/shared/{period}/{share_type}.json
> - Action:call ...
> ```
>
> We will include additional explanations regarding the prompt format in the revised paper.
>
> Given that our experiment was conducted using ReAct, we consider our experiment setting to be already relying on chain-of-thought.
>
> For open-source LLM baselines, we presented the performance of more open-source LLMs (Codellama, Qwen) in the table of A1.

---

> ### Author Response · Authors · 2024-11-20
>
> **Q4.** What was the setting for the GPT-based experiments - zero-shot or in-context learning? Please add these details in the next version of the draft
>
> **A4.** Thank you for your suggestion. The GPT-based experiments used zero-shot prompting. We will clarify this in the revision.

---

> ### Author Response · Authors · 2024-11-20
>
> **Q5.** Could you explain why does GPT-4o-mini perform better without ground truth in Table 4?
>
> **A5.** Error analysis shows that GPT-4o-mini in the With GT setting more frequently misjudged API confirmation status compared to the W/O GT setting.
>
> The following table shows a summary of our error analysis on DST across the models.
>
> **DST Error analysis**
> |     | GPT-3.5-turbo |   | GPT-4o-mini |   | GPT-4-turbo | | GPT-4o |         | Llama3-8b-inst |        | TD-Llama    |
> |----|----|-----|---|---|---|--|----|---|-----|----|---|
> | | W GT | W/O GT | W GT | W/O GT  | W GT| W/O GT | W GT| W/O GT | W GT | W/O GT| W GT| W/O GT |
> | # of Error  | 4128    | 4512   | **2781**  | **2177**   | 1515  | 2117   | 1257 | 2169   | 3138| 5090   | 492  | 1873|
> | Generation Error    | 0 | 0 | **0** | **0** | 0 | 0| 0| 0| 3| 0| 260| 1619|
> | API Conf. Err (GT = T) | 1609 | 1841| **1060** | **504** | 211| 224 | 243| 1607| 583| 1014 | 30   | 1 |
> | API Conf. Err (GT = F)| 750  | 410 | **848**| **373**| 692| 891 | 343 | 133 | 723| 923| 0 | 0 |
> | Format Err| 532| 502| **153**| **0**| 0| 0| 74| 0| 531| 319| 61|103|
> | Slot Error| 1139 | 1674| **508**| **912** | 430| 774 | 443| 221| 1101| 2663| 6| 23|
> | Value Error |561| 630| **398**|**823**| 498| 626| 495| 221| 846| 1423| 134| 144|
>
> - The W/O GT results for TD-Llama could not be fully entered due to length constraints, so we will present here in text form. From Generation Error to Value Error, the results are 1873, 1619, 1, 0, 103, 23, and 144, respectively.)
>
> - The DST error table shows error counts, where slot and value errors can overlap in a single prediction, causing their sum to exceed the total errors.
>
> API Conf. Err (GT = F) refers to an error where the system incorrectly determines that the API has been confirmed when it has not, and API Conf. Err (GT = T) is the reverse. While other models show higher errors in the W/O GT setting, GPT-4o-mini does not follow this trend. This is the main reason why GPT-4o-mini performs better in the W/O GT setting compared to the With GT setting.
>
> For better understanding, we have provided a description of each error type on **Error Type**. By referring to the **Format of the dialogue state**, you will be able to better understand exactly which aspects each error type pertains to.
>
> **Format of the dialogue state** (More details are in Appendix.4 of the paper)
>
> - When there is no confirmed API:
>
>   - Dialogue State: {API confirmed: false, API status: none}
>
> - When the API is confirmed and some input parameter information can be extracted from dialogue history:
>
>   - Dialogue State: {API confirmed: true, API status: {API name: “API1”, Required parameters: {param1: “value1”, param2: “”}, Optional parameters: {param3: “value3”}}}
>
> Here, we call the **param1, param2, param3 as slots**, and **value1, value3 as values.**
>
> **Error Type**
>
> - **Generation Error**: This occurs when the dialogue state dictionary is not generated at all.
>
> - **API Confirmation Error (GT = True)**: This error happens when API is confirmed (api_confirmed=true), but is incorrectly predicted as it is confirmed (api_confirmed=false).
>
> - **API Confirmation Error (GT = False)**: This error occurs when the API is not confirmed (api_confirmed=false), but the model incorrectly predicts it as confirmed (api_confirmed=true).
>
> - **Format Error**: This occurs when the dialogue state does not fully generate all fields such as api_confirmed, api_status, required parameters, and optional parameters.
>
> - **Slot Error**: When api_confirmed is true, this error involves generating a dialogue state that does not include all required and optional parameter slots as specified in the API documentation.
>
> - **Value Error**: This error involves incorrectly extracting the slot’s value from the dialogue history, with the following types:
>
>   - **Extracting Input Value from Multiple Result Error**: This error occurs when an appropriate value cannot be selected from multiple results returned by the API output (as seen in turns 6 and 7 of Figure 2).
>
>   - **Inform Intent Add Error**: This occurs when there is a value within the user query that could be used as an input parameter (inform intent clear add), but the model fails to track it.
>
>    - **Other General Input Parameter Extraction Errors**: Errors that occur in typical situations where the input parameter is extracted incorrectly.
>
> **Result analysis for DST error**
> - Various errors appear in the GPT series and Llama3 Instruct models. In the W/O GT setting, which resembles real-world scenarios, more errors occur.
> - In TD-Llama, errors decrease significantly, though generation errors remain common. Other frequent errors include Extracting Input Value from Multiple Results and Inform Intent Add errors.
> - To reduce errors through fine-tuning, a simple approach to address "generation errors" can be proposed. However, "Extracting Input Value from Multiple Result Error," involving noisy lists, remains a challenging problem and an important future direction.

---

> ### Author Response · Authors · 2024-11-20
>
> **Q6.** What are the domains of the examples in the dataset?
>
> **A6.** ToolDial includes the following 28 domains:
>
> _['Business', 'Data', 'Energy', 'Finance', 'Food', 'Business_Software', 'Database', 'Entertainment', 'Events', 'Health_and_Fitness', 'Location', 'SMS', 'Science', 'Social', 'Mapping', 'Music', 'Search', 'Logistics', 'Media', 'News_Media', 'Sports', 'Travel', 'Weather', 'Transportation', 'Financial', 'Gaming', 'Payments', 'Commerce']_
>
> The proportions of the domains are as follows:
> - Data: 21.591%
> - Sports: 18.702%
> - Weather: 16.020%
> - Media: 10.143%
> - Location: 6.975%
>
> These five domains make up the majority, with the remaining 23 domains evenly distributed.

---

> ### Author Response · Authors · 2024-11-20
>
> **Sugg1.** Please mention the language of the dataset in the paper
>
> **A1.** Yes, we will mention that the dataset is in English in the Dialogue Generation section (Section 3.4).

---

> ### Author Response · Authors · 2024-11-20
>
> **Sugg2.** Please include qualitative examples from the dataset describing (i) examples after the data collection is complete and (ii) failures of respective LLMs
>
> **A2.** Yes, we will add examples of the dataset after the data collection is complete and error cases of each LLM in the paper.

---

> ### Author Response · Authors · 2024-11-20
>
> **Sugg3.** Please report at least three significant digits when reporting the results
>
> **A3.** Yes, we will update the numeric format in all tables accordingly.

---

> ### Author Response · Authors · 2024-11-20
>
> **Sugg4.** As it is a multi-turn dataset, inclusion of task completion metric - when the model performs all the correct api calls at every turn it is marked as a complete task.
>
> **A4.** Thank you for your suggestion. After each user utterance, we ensure that the system generates the appropriate dialogue state. Our DST experiment evaluates this accuracy for every turn. Therefore, if the goal is to measure whether the correct API can be called at "every turn", the DST score we reported in the experimental results reflects this measurement.
>
> If we focus only on the turns where an API is actually called, DST scores are as follows.
> |                   | With GT | W/O GT |
> |-------------------|---------|--------|
> | GPT-3.5-turbo     | 45.1    | 22.1   |
> | GPT-4o-mini       | 78.3    | 78.9   |
> | GPT-4-turbo       | 87.8    | 81.4   |
> | GPT-4o            | 88.9    | 84.2   |
> | Llama3-8b-Instruct| 81.0    | 10.7   |
> | TD-Llama          | 90.4    | 67.4   |
>
> Please note that the Llama scores have been updated as explained in A1.

---

> ### Author Response · Authors · 2024-11-20
>
> **Sugg5.** Please include vanilla LLaMa results in Table 5.
>
> **A5.** Thank you for your suggestion. We have added the scores of vanilla Llama as follows and we will update Table 5 in our revision accordingly.
>
> **F1-score table**
> | Setting | Model               | response | responsefail | request | retrievecall | clarify | systembye | suggest | call |
> |---------|---------------------|----------|--------------|---------|--------------|---------|-----------|---------|------|
> | **With GT**    | GPT-3.5-turbo              | 63.8     | 0.0          | 28.4    | 66.2         | 1.3     | 95.5      | 0.0     | 53.4 |
> |         | GPT-4o-mini          | 78.9     | 0.0          | 44.3    | 67.4         | 64.5    | 97.2      | 0.0     | 67.0 |
> |         | GPT-4-turbo        | 93.6     | 0.0          | 18.1    | 87.5         | 56.7    | 97.2      | 29.9    | 56.4 |
> |         | GPT-4o             | 88.3     | 0.0          | 13.7    | 74.9         | 29.6    | 97.2      | 24.6    | 54.1 |
> |         | Llama3-8b-Instruct | 46.4     | 0.0          | 8.5     | 23.7         | 0.0     | 99.8      | 14.0    | 44.4 |
> |         | TD-Llama           | 100.0    | 77.5         | 44.8    | 97.2         | 77.4    | 99.9      | 16.8    | 68.6 |
> | **W/O GT**   | GPT-3.5-turbo              | 70.7     | 0.0          | 1.3     | 77.6         | 0.0     | 93.0      | 0.0     | 49.7 |
> |         | GPT-4o-mini          | 88.5     | 0.0          | 36.1    | 62.6         | 0.0     | 97.2      | 0.0     | 65.1 |
> |         | GPT-4-turbo        | 96.6     | 0.0          | 10.8    | 79.9         | 40.6    | 97.2      | 35.5    | 57.8 |
> |         | GPT-4o             | 95.8     | 0.0          | 14.3    | 81.2         | 38.6    | 97.2      | 46.1    | 62.0 |
> |         | Llama3-8b-Instruct | 30.5     | 0.0          | 1.9     | 27.3         | 0.0     | 93.1      | 9.4     | 42.0 |
> |         | TD-Llama           | 98.2     | 99.1         | 78.4    | 94.5         | 99.8    | 100.0     | 99.9    | 86.9 |

---

> ### Author Response · Authors · 2024-11-20
>
> **Sugg6.** While GPT based results are not reproducible (based on API changes), it may be useful to release the predictions from those runs for future work to be built on this work.
>
> **A6.** Thank you for your suggestion. We will include the GPT predictions along as part of our code release.

---

> ### Author Response · Authors · 2024-11-20
>
> **Sugg7.** While it is exciting to benchmark the models on, non LLM based baselines such as BERT based/T5 based dialogue state tracking/action recognition should also be evaluated for completeness
>
> **A7.** Thank you for your suggestion. For dialogue state tracking and action prediction, LLM-based models (such as SimpleTOD [1]) have proven to be superior to non-LLM models. We believe that it is already well-established in the literature that LLM models generally outperform non-LLM models, which is why we did not include non-LLM baselines in our experiments. However, if you believe it is still essential to include these baselines, we will incorporate their results in the revision.
>
> [1] HOSSEINI-ASL, Ehsan, et al. A simple language model for task-oriented dialogue. Advances in Neural Information Processing Systems, 2020, 33: 20179-20191.

---

> ### Author Response · Authors · 2024-11-20
>
> **Sugg8.** The dataset construction section is a little hard to understand. Perhaps a running example may make the section clear.
>
> **A8.** Using a running example would be helpful in clarifying the data construction part. We will enhance Section 3.2 by incorporating the following example based on Figure 2.
>
> **3.2 Action Sequence**
>
> One triple consisting of two nodes was sampled from the graph. Suppose this triple includes the APIs "LeagueHomeStandings" and "CategoryTournaments". LeagueHomeStandings requires two input parameters, CategoryTournaments typically returns multiple IDs as output, and one of the input parameters of LeagueHomeStandings, tournamentId, and the output component id of CategoryTournaments are connected by an edge (see the reasoning step in the 6th turn of Figure 2).
>
> From this triple, possible action sequences include:
>
> - inform intent clear → retriever call → request → fail inform → retriever call → request → inform → call → request → inform → call → response
> - inform intent vague → clarify → inform intent clear → retriever call → request → fail inform → retriever call → request → inform → call → request → inform → call → response
> - inform intent vague → suggest → affirm → request → fail inform → retriever call → request → inform → call → request → inform → call → response
> - inform intent vague → suggest → negate → response fail
> - And more
>
> To enhance the diversity of conversational flows in our dataset, we generate a dialogue for each of these sequences. For instance, the dialogue in Figure 2 is based on the first action sequence above.
>
> There are three request actions in this action sequence. The first request is for the input parameters needed to execute LeagueHomeStandings, the second is to execute CategoryTournaments, and the third is to select one ID from the multiple IDs output by CategoryTournaments. In Figure 2, CategoryTournaments is an API that returns multiple IDs, requiring an additional user prompt to select the correct value (see the 6th turn of Figure 2).
>
> **3.3 Dialogue scenario generation**
>
> During the dialogue scenario generation stage, we plan what information each user and system utterance should include in each turn  to perform the corresponding action (e.g., input parameters the system would request, parameter values the user would provide, etc.).
> The dialogue in Figure 2 is generated from a prompt like the following:
>
> - Inform intent clear: Instruction: the user utters a pre-constructed query related with API LeagueHomeStandings and CategoryTournament.
> - (Retriever call) -> request: Instruction: the system to ask the user for seasonId and tournamentId as input parameters.
> - Fail inform: Instruction: the user responds with seasonId ‘45’ but fails to provide tournamentId.
> - (Retriever call) -> request: Instruction: the system prompts the user for id.
> - Inform: Instruction: the user responds with the requested information.
> - (call) -> request: Instruction: the system asks the user for the name variable, to select one Id from multiple results.
> - Inform: Instruction: the user responds with "NBA."
> - (call) -> response: Instruction: the system responds based on the results of the call.
>
> For parameter values (e.g., 45, 264 in Figure 2), plausible inputs and outputs for the API used in dialogue generation are first generated and then utilized.
>
> Scenario prompts are created by predefining templates for each action and filling them with the API's input parameters and values. During this process, the dialogue states used in the experiment are also automatically generated for each turn, using the same values. The prompts and dialogue states are generated by executing pre-written code.
>
> The scenario prompt generated in this way is then provided to GPT-4o, along with simple instructions, to generate the dialogue data (in Section 3.4).

---

> ### Author Response · Authors · 2024-11-20
>
> **Sugg9.** [Future work] As API calls are similar to function calling in Python, it would be useful to also benchmark CodeLLaMa/finet-uned CodeLLaMa for the given benchmark
>
> **A9.** Thank you for your suggestion. We have added the scores of Code Llama in the table in A1.

---

> ### Author Response · Authors · 2024-11-20
>
> And for the missing reference, thank you for sharing. We will discuss these works in our revision.

---

> ### Author Response · Authors · 2024-11-22
>
> Thank you again for your time and effort to review our paper and provide constructive feedback. We have made our best effort to incorporate your suggestions and address your questions in our rebuttal. We understand you're busy but we would greatly appreciate it if you could review our rebuttal and let us know if it addresses your feedback and concerns.

---

### Author Response · Authors · 2024-11-24
**Review Summary**

Dear Area Chair and Reviewers,

We sincerely appreciate the time and effort you have dedicated to thoroughly reviewing our paper. The following summarizes the contributions of our work as highlighted by the reviewers:

1. The ToolDial dataset addresses a critical gap in TALM literature by providing extensive, nuanced multi-turn dialogue interactions that facilitate timely exploration of tool usage in real-world scenarios (Reviewers fu86, 4sGc).

2. The benchmark's smaller, focused tasks enable a comprehensive evaluation of various language models on ToolDial, offering actionable insights into the challenges TALMs face in multi-turn, tool-augmented interactions (Reviewers 4sGc, fu86).

3. The dataset is generated using an API graph with GPT models, reducing reliance on human-curated data and offering a scalable and inspiring approach that could inform future research (Reviewer fu86).

4. Our paper establishes a timely new benchmark that emphasizes multi-turn interactions, distinguishing it from older API benchmarks like API Bank, while addressing gaps left by aging datasets such as MWOZ and SGD. As a result, the paper makes solid contributions to the dialogue community (Reviewer qYBo).

The reviewers have also pointed out several weaknesses, which we have carefully addressed through detailed responses and supporting experiments and analyses. Those include internal biases in GPT during dialogue generation, additional baselines and error analysis, unclarity in the writing, and some missing related work. We have revised the paper to address these concerns and enhance the clarity.

Once again, we deeply appreciate the constructive feedback provided, which has greatly advanced our research.

---

### Comment · Area_Chair_3v2X · 2024-11-25
**Action Required: Respond to Author Rebuttals - Nov 27**

Dear ICLR Reviewers,

The author discussion phase is ending soon. Please promptly review and respond to author rebuttals for your assigned papers. Your engagement is critical for the decision-making process.

Deadlines:
- November 26: Last day for reviewers to ask questions to authors.
- November 27: Last day for authors to respond to reviewers.
- November 28 - December 10: Reviewer and area chair discussion phase.

Thank you for your timely attention to this matter.

---

### Meta-Review · Area_Chair_3v2X · 2024-12-21

**Metareview:**

The paper introduces ToolDial, a new dataset for evaluating Tool-Augmented Language Models (TALMs) in multi-turn dialogue settings. The dataset contains 11,111 dialogues with an average of 8.95 turns, featuring complex interactions where API calls depend on context from previous turns and may require requesting additional information from users.

The reviewers value the timeliness and importance of addressing multi-turn tool calling capabilities, as well as the comprehensive evaluation across different tasks. However, reviewers are still a bit concerned about LLM-generated data quality,  evaluation metrics, and missing detailed error analysis.

Overall, I recommend acceptance given the paper's contribution to tool learning research.

**Additional Comments On Reviewer Discussion:**

Already discussed in above.

---

### Decision · Program_Chairs · 2025-01-22

Accept (Poster)